# Study on the Synthesis, Antioxidant Properties, and Self-Assembly of Carotenoid–Flavonoid Conjugates

**DOI:** 10.3390/molecules25030636

**Published:** 2020-02-01

**Authors:** Ildikó Línzembold, Dalma Czett, Katalin Böddi, Tibor Kurtán, Sándor Balázs Király, Gergely Gulyás-Fekete, Anikó Takátsy, Tamás Lóránd, József Deli, Attila Agócs, Veronika Nagy

**Affiliations:** 1Department of Biochemistry and Medical Chemistry, University of Pécs, Medical School, Szigeti út 12, H-7624 Pécs, Hungary; ildiko.szabo@aok.pte.hu (I.L.); czett.dalma@gmail.com (D.C.); Katalin.Boddi@aok.pte.hu (K.B.); gergely.gulyas@aok.pte.hu (G.G.-F.); Aniko.Takatsy@aok.pte.hu (A.T.); Tamas.Lorand@aok.pte.hu (T.L.); Jozsef.Deli@aok.pte.hu (J.D.); attila.agocs@aok.pte.hu (A.A.); 2Department of Organic Chemistry, University of Debrecen, POB 400, H-4002 Debrecen, Hungary; kurtan.tibor@science.unideb.hu (T.K.); kiraly.sandor.balazs@science.unideb.hu (S.B.K.); 3Department of Pharmacognosy, University of Pécs, Faculty of Pharmacy, Rókus u. 2, H-7624 Pécs, Hungary

**Keywords:** carotenoids, flavonoids, click-reaction, antioxidant capacity, supramolecular chirality, zeaxanthin–flavonoid conjugates, electronic circular dichroism

## Abstract

Flavonoids and carotenoids possess beneficial physiological effects, such as high antioxidant capacity, anticarcinogenic, immunomodulatory, and anti-inflammatory properties, as well as protective effects against UV light. The covalent coupling of hydrophobic carotenoids with hydrophilic flavonoids, such as daidzein and chrysin, was achieved, resulting in new amphipathic structures. 7-Azidohexyl ethers of daidzein and chrysin were prepared in five steps, and their azide-alkyne [4 + 2] cycloaddition with pentynoates of 8′-apo-β-carotenol, zeaxanthin, and capsanthin afforded carotenoid–flavonoid conjugates. The trolox-equivalent antioxidant capacity against ABTS^•+^ radical cation and self-assembly of the final products were examined. The 1:1 flavonoid–carotenoid hybrids generally showed higher antioxidant activity than their parent flavonoids but lower than that of the corresponding carotenoids. The diflavonoid hybrids of zeaxanthin and capsanthin, however, were found to exhibit a synergistic enhancement in antioxidant capacities. ECD (electronic circular dichroism) and UV-vis analysis of zeaxanthin–flavonoid conjugates revealed that they form different optically active J-aggregates in acetone/water and tetrahydrofuran/water mixtures depending on the solvent ratio and type of the applied aprotic polar solvent, while the capsanthin derivatives showed no self-assembly. The zeaxanthin bis-triazole conjugates with daidzein and with chrysin, differing only in the position of a phenolic hydroxyl group, showed significantly different aggregation profile upon the addition of water.

## 1. Introduction

Countless natural bioactive compounds have been isolated from food products and their effects on human health were studied. However, an increasing amount of evidence suggests that a nutrient may have a specific action, the combination of nutrients displays different (even stronger) effects than a sole component [1,2]. As foods are complex matrices, the bioaccessibility and bioavailability of a component are obviously influenced by the presence or absence of other constituents [3,4]. At the same time, significant synergy was found in the physiological effects of many small molecules used in combination [5].

Our attention was directed to two natural antioxidant families, carotenoids and flavonoids. Both are abundant in human nutrition and are connected to a beneficial influence on health. Carotenoids are hydrophobic pigments that reduce the risks of a number of chronic diseases [6]. Especially, lutein and zeaxanthin play an important role in the prevention of age-related macular degeneration, and were found to be useful in the treatment of neurodegenerative disorders, such as Alzheimer’s disease [7].

Flavonoids are hydrophilic compounds and occur frequently in the form of glycosides. They are referred to as multipotent antioxidants, as they have other favorable pharmacological effects [8]. Daidzein, an isoflavone, was found to be advantageous in cardiovascular diseases [9], bone resorption [10], and cell proliferation, but some adverse effects were also described [11]. Chrysin is a flavone with low bioavailability and rapid excretion; however, it showed cancer chemopreventive activity [12], decreased the symptoms of colitis [13], and was also found to be beneficial in connection with the insulin resistance and type II diabetes mellitus [14].

Both carotenoids and flavonoids are rather sensitive to oxidation, but their stability and bioavailability can be significantly increased by additives. Water-soluble carbohydrate-based nanoparticles were described to form host–guest complexes with carotenoids, improving their properties [15]. The effect of mixing carotenoids with flavonoids is of particular interest, as synergy arises in their protective effect on low-density lipoprotein oxidation, which plays an important role in the development of atherosclerosis [16]. Daidzein was found to regenerate carotenoids from their radical cation forms [17]. The carotenoid–flavonoid interactions, which happen at the water/lipid interface, were found to be of importance for the function of both antioxidants [18].

The covalent coupling of these antioxidants with molecules of opposite polarity opens new perspectives, as the amphiphilic structures allow an unusual location for the antioxidant molecules in cells, and they can exert their effects in a broader range. Such modifications were found to also alter the activity of the molecules in vitro experiments. Joining carotenoids to hydrophilic molecules resulted in better stability and solubility, as well as increased antioxidant activity [19]. The esterification of the flavonoid silybin with lipophilic fatty acids ameliorated its antioxidant and antiviral properties [20]. The covalent conjugation of silybin with l-ascorbic acid, trolox alcohol, or tyrosol through a lipophilic linker provided an efficient protective action against lipid peroxidation by modifying the electron transfer abilities of the connecting moieties [21].

There are some examples for the synthesis of flavonoid–carotenoid hybrids, which couple a shorter than C_40_ apocarotenoid moiety to flavonoids directly, without any spacer. Terpenoid C_25_ or C_30_ aldehydes were connected to hydroxylated or non-hydroxylated 6-(diethoxyphosphoryl) methylflavones by the Wittig–Horner–Emmons reaction [22]. The resulting continuously conjugated carotenyl flavonoids showed improved photoprotective activities, and they surpassed the individual constituents in their antioxidant properties [22,23].

When 14′-apo-β-carotenoic acid was conjugated with daidzein in position 3b by a C-C bond, allowing the formation of a continuously conjugated π-system, the product exhibited a significantly better protective effect against lipooxidation in membranes compared to the parent compounds alone or in combination [24].

Daidzein (in position 7a) and quercetin (in positions 3b and 4b) were esterified by 14′-apo-β-carotenoic acid using 4-dimethylaminopyridine (DMAP)/*N*-ethyl-*N*′-(3-dimethylaminopropyl) carbodiimide (EDC) coupling. Interestingly, the antioxidant activity of daidzein increased while that of quercetin decreased after derivatization [25].

A similar DCC or EDC/DMAP esterification method was applied for the synthesis of retinoate esters of various flavonolignans in low yields. The prepared esters showed increased radical scavenging activity [26].

Here, we present a straightforward method for the regioselective covalent coupling of daidzein and chrysin with various carotenoids through a long spacer, which ensures flexibility and allows the hydrophilic and lipophilic moieties to move away or draw closer. Three different types of carotenoids were studied, the C_30_ primary alcohol 8′-apo-β-carotenol, the symmetric secondary diol zeaxanthin, and the asymmetric κ-carotenoid capsanthin. The antioxidant activity and the supramolecular organization of the prepared amphipathic compounds were also investigated.

## 2. Results and Discussion

### 2.1. Synthesis

Both carotenoids and flavonoids are sensitive compounds, and for the coupling of these molecules, a very mild method was needed. Previously, azide-alkyne [4 + 2] cycloaddition (click-reaction) was successfully applied in the synthesis of carotenoid conjugates with polyethylene glycols [27] or with carbohydrates [28]. Since the pentynoate esters of carotenoids can be easily prepared [28], we worked out a procedure for the synthesis of azido-derivatives of flavonoids.

The isoflavone daidzein (**1**) and the flavone chrysin (**7**) were the molecules of choice for the preparation of 7-azidohexyl ether derivatives in five steps (Figure 1). The flavonoids (**1**,**7**) were acetylated by acetic anhydride in pyridine, and their position 7 were selectively deprotected using imidazole [29]. A bromohexyl moiety was introduced in position 7 by the Mitsonobu reaction, and after complete deacetylation, an azide substitution was applied.

The 7-azidohexyl ether of daidzein and chrysin (**6**,**12**) was reacted with various carotenoid pentynoates (8′-apo-β-carotenol-pentynoate **13**, zeaxanthin dipentynoate **14**, capsanthin dipentynoate **15**) in the presence of bis-triphenylphosphano-copper(I)-butyrate complex [30] as a catalyst, in dichloromethane. (Figure 2). The azido-flavonoids were applied in small excess (1.3–1.5 eq. per triple bond), providing the corresponding triazoles in satisfactory or good yields (60%–80%). In the case of the 8′-apo-β-carotenol-pentynoate **13**, only monotriazoles can be formed. The other carotenoids were used as dipentynoate esters, and during the click-reaction the formation of monotriazoles was also observed. When the reactions were followed by TLC, the monotriazoles appeared first, and in a longer reaction time, they were further converted to bistriazole derivatives. However, small amounts of monotriazoles always remained at the end of the reactions, even with a higher excess of azidohexyl ethers. Yields for the monotriazoles were calculated from the starting carotenoid pentynoates, as if they were the sole products. The capsanthin pentynoate (**15**) did not give a complete conversion, as approximately 10% of the starting pentynoate was recovered.

All the prepared compounds were fully characterized by their NMR, MS, UV, and IR spectra. As capsanthin is a non-symmetric carotenoid, it can provide two regioisomeric monotriazole derivatives. In the case of the daidzein-capsanthin conjugates (**20a** and **20b**), these regioisomers could be separated and identified. In the ^1^H-NMR spectra of **20a** and **20b**, the chemical shifts of H-3′ are slightly but characteristically different (Table 1), and in comparison with the parent capsanthin dipentynoate (**15**), as well as with the bistriazole **19**, the substituents on C-3′ can be distinguished. Similar differences in the chemical shifts of H-3 can also be observed.

The two regioisomeric monotriazoles of **25** were also observed on TLC, but since they were formed in low yields and their separation seemed to be extremely challenging, we used them as a mixture for further investigations.

### 2.2. Measurement of Antioxidant Capacity

The trolox-equivalent antioxidant capacity (TEAC) of the prepared flavonoid–carotenoid conjugates (**16**–**25**) was investigated towards ABTS^•+^ radical cation in aqueous ethanolic solution (see the experimental section) [31]. The synthetized compounds showed significantly higher TEAC values than their parent flavonoids (Figure 3, Table 2); the exceptions were the conjugates of 8′-apo-β-carotenol (**16**,**21**), which showed a similar activity to daidzein or chrysin.

Zeaxanthin conjugated with one daidzein (**18**) or chrysin (**23**) was found to be weaker antioxidants than zeaxanthin; however, its bistriazole derivatives (**17** and **22**) exceeded the parent zeaxanthin. The antioxidant capacity of capsanthin did not significantly change on conjugation with one daidzein, and no differences in the TEAC values of monotriazoles **20a** and **20b** were found. The bistriazole derivative **19**, however, surpassed capsanthin in antioxidant capacity. The conjugation with chrysin lowered the TEAC value in small extents. The 8′-apo-β-carotenol was found to be a better antioxidant than its flavonoid derivatives.

The daidzein-bistriazole derivatives of zeaxanthin and capsanthin were the best antioxidants under the examined conditions. The TEAC value of a simple zeaxanthin:daidzein 1:2 mixture is much lower than that of the bistriazole conjugate **17**, which indicates a synergistic enhancement in antioxidant capacity.

The above results were obtained in ethanolic solution of ABTS^•+^ (see the experimental section). The antioxidant property of compound **17** was also determined in aqueous ABTS^•+^ solution, where the TEAC value was found to drop by ca. 20%. The reason for this phenomenon can be explained by the water-induced aggregation of the carotenoid derivative, which is an accordance with literature data [32].

### 2.3. Supramolecular Assembly

When the coupling click-reactions were followed by TLC, and samples were taken directly from the dichloromethane reaction mixtures, the formation of an intense orange-red spot on the start-line was observed in the hexane-acetone eluent while the spots of the products remained faint. This phenomenon generally suggests a salt formation; however, it was hardly possible in these reactions. After the disappearance of the starting materials, the dichloromethane solvent was evaporated from the reaction mixture of 8′-apo-β-carotenol-pentynoate and 7-azidohexyl daidzein, then the reaction mixture was taken up in acetone. Interestingly, when the acetonic solution was applied on the TLC plate and eluted by the same eluent (hexane-acetone), no start-point substance remained, but the whole amount was eluted with the R_f_ value of the product. This finding was rather unusual, as the retention properties of a substance depend on the quality/polarity of the eluent but independent of the solvent in which it is applied on the plate, since the plate is dried before elution. We repeated the TLC experiment using different solvents for the application of the sample, but the same hexane-acetone eluent for the development. The product triazole **16** in methanolic, chloroformic, or ethyl acetate solutions was found to behave similarly as in dichloromethane, i.e., it gave a start-spot, while it was eluted when it was applied in acetonic solution. From these observations, we suspected the formation of some aggregates, which formed in the presence of certain solvents on drying and did not moved on the chromatographic plate.

Carotenoids are known to form self-assemblies among certain conditions even in nature. These supramolecular organizations frequently have ordered structures (H- or J-type aggregates), which results in significant changes in the UV-Vis spectra of carotenoids [33,34,35]. Moreover, the aggregates can exhibit a so-called supramolecular exciton chirality due to the optically active helical structures, which can be studied by electronic circular dichroism (ECD) spectroscopy [36,37]. Aggregation of amphipathic carotenoid derivatives was also examined thoroughly [38].

As **16** daidzein-8′-apo-β-carotenol hybrid is not chiral, its aggregation can only be examined by the change in its UV-Vis properties. The absorption spectrum of the ethanolic solution of **16** upon aqueous dilution showed an intense hypsochromic and a small bathochromic shift that suggest self-assembly of **16** with the formation of J-aggregate (Appendix A).

The aggregation properties of the optically active bistriazoles **17**, **19**, **22**, and **24**, as well as monotriazole **23** were investigated by ECD spectroscopy. The acetone or tetrahydrofuran (THF) solution of these derivatives were diluted with water, and the ECD spectra were recorded with the different dilutions. Upon dilution with water, the optically active capsanthin derivatives (**19**, **24**) did not show any significant increase in the ECD transitions that would have suggested supramolecular organization. In contrast, intense ECD bands appeared in the spectra of the bistriazole daidzein-zeaxanthin conjugate **17** in acetone when the water content was increased above 25% (Figure 4a).

The absorption spectra showed a characteristic bathochromic shift with a new band at 519 nm, which indicated the formation of a weakly-coupled J-aggregate (Appendix A). Zeaxanthin and zeaxanthin derivatives modified at the *sec*-hydroxyl group can form both J- and H-aggregates depending on the applied pH, initial concentration, and solvent/water ratio [39,40]. The intense 480 nm positive cotton effect (CE) with shoulders at 456 and 519 nm, the broad and intense negative CE at 412 nm, and the weak negative CE at 547 nm are characteristic of a J-aggregate also observed in the EtOH/water solution of zeaxanthin under certain conditions [39,40]. Interestingly, a similar ECD profile was observed in sergeant-soldier aggregate of 5% α- and 95% β-carotene [37], in which weak van der Waals interactions also prevail. The ECD spectra recorded in the range of 1:0–1:3 acetone/water ratio showed strong characteristic ECD bands from a 2:1 ratio and the largest amplitudes were observed at a 1:2 ratio. A further increase of the water content to a 1:3 ratio decreased both the ECD and absorption intensity. The intensities of the five CEs did not show a monotonous change with the increase of the water content; the 547 nm CE had maximum intensity at the 1:1 ratio, while the 480 and 412 CEs had the largest amplitude at the 1:2 ratio.

In THF, a similar ECD feature appeared with the increase of the water ratio, and the maximum ECD intensity was reached at the 1:5 ratio, while a further increase of the water content decreased the amplitude of the CEs (Figure 4b). The absorption and ECD spectra of **17** at the 2:3 and 1:2 ratio differed from those at the 1:3, 1:5, and 1:10 ratios, which indicated the presence of at least two different J-aggregates.

The bistriazole chrysin-zeaxanthin conjugate (**22**) also formed a J-aggregate with the increasing amount of water in acetone, which was, however, different from that of **17**. The conjugate **22** had a bisignate ECD curve even in pure acetone, indicating aggregate formation in the absence of water (Figure 5a,b). In acetone/water 5:1, the high-wavelength positive CE shifted to the red with increased intensity. The 545 nm positive CE with shoulders at 508 and 464 nm and the negative CE at 397 nm represented an ECD pattern similar to that of **17**, with a considerable red shift. In acetone/water 4:1, the intensity increased significantly and reached the maximum, while further addition of water flattened the ECD curve and changed the absorption maxima. The diminishing ECD intensity in acetone/water ratio 2:1–1:3 suggested the deterioration of the J-aggregate.

In the THF solution, **22** had a low-intensity ECD spectrum (Figure 5c,d), different from that in acetone, which did not change much up to the 2:3 THF/water ratio. At the 1:2 THF/water ratio, an intense negative ECD couplet centered at 425 nm appeared, which was red shifted with the 1:3 ratio. A further significant change occurred at the 1:5 and 1:10 ratios, the ECD curves of which differed only in the amplitudes and they had a similar ECD pattern to that of **17**, with opposite signs of CEs. The ECD spectra of **22** at the high water ratio were very different from those in acetone, and the mirror-image relationship with the ECD spectrum of **17** acetone/water 1:2 indicated their opposite sense of supramolecular chirality. The three different ECD and absorption spectra suggested the presence of at least three different types of J-aggregate.

The zeaxanthin conjugates **17** and **22** differ only in the attached daidzein/chrysin flavonoid moiety, but they produced markedly different aggregation profiles upon the addition of water in both acetone and THF. The aggregate formation was also sensitive to the type of non-protic solvent, acetone versus THF.

In the acetone/water mixtures, the monotriazole chrysin–zeaxanthin conjugate (**23**) showed the same characteristic changes in both the ECD and absorption spectra as the bistriazole daidzein–zeaxanthin conjugate **17**, indicating the formation of the same J-aggregate (Figure 6a,b).

The ECD transitions above 400 nm reached the maximum intensities at the 1:1 ratio and there was no significant decrease in the intensity up to the 1:3 ratio.

In the THF/water mixtures, a completely different aggregation process could be observed (Figure 7a,b). Aggregation started at the 1:1 ratio and the negative ECD couplet reached the maximum intensity at the 1:2 ratio. The intensity of the ECD bands was so high at the initial concentration that a sixfold dilution had to be applied to record the ECD bands properly. According to the characteristic 530 nm absorption band, a J-aggregate formed at the 2:3 and 1:2 ratios.

By increasing the ratio of water further, the negative couplet was inverted to a positive one at the 1:5 and 1:10 ratios. At these ratios, both hypsochromic and bathochromic shifts of the absorption bands took place compared to the bands at the 1:1 ratio (Figure 8). The inversion of the ECD couplet with the increasing amount of water in THF suggests the inversion of chirality in the aggregate, which could not be observed in the acetone/water mixtures.

## 3. Conclusions

A straightforward method was worked out for the synthesis of novel carotenoid–flavonoid conjugates and their antioxidant capacities were measured. Some of these amphipathic compounds containing a flexible linker were found to show self-aggregation in the presence of water. The antioxidant capacity was measured in ethanolic solutions. Determination of the TEAC values in aqueous medium seems to be more relevant biologically, but most probably, the antioxidant property under such conditions rather belongs to the aggregates than to the individual molecules. A further study is needed to clarify how the aggregation influences the antioxidant capacity. In the future, the solvent dependence of the antioxidant activity, and the structure of the aggregates will be examined with atomic force microscopy.

## 4. Experimental Section

### 4.1. Materials and Methods

Melting points were measured on a Stuart SMP30 apparatus (Cole-Parmer, Staffordshire, UK) and are uncorrected. The UV spectra were implemented on a Jasco spectrophotometer model V-550 UV/Vis (JASCO, Tokyo, Japan). ECD spectra were recorded at room temperature with a J-810 spectropolarimeter (JASCO, Tokyo, Japan). The IR spectra were run on an Impact 400 FT-IR spectrophotometer (Nicolet, Germany) in KBr pellets using a KBr pellet as the background reference spectrum. NMR spectra were recorded with a Bruker Avance III Ascend 500 spectrometer (500/125 MHz for ^1^H/^13^C) (Bruker BioSpin GmbH, Rheinstetten, Germany). The ^13^C and ^1^H-NMR assignments for **6**, **12**, **16**, **17**, **19**, and **23** were made on the basis of 1D (^1^H, ^13^C APT) and 2D (HSQC, HMBC) experiments, and the assignments for the other compounds were based on structural similarities with the above molecules. Chemical shifts were referenced to the residual solvent signals, or to Me_4_Si (^1^H). A special numbering of H and C atoms, indicated in the formulas (Figure 9), for the flavonoid moieties was applied, since the structure of the final coupled products was rather complex.

Molar masses were obtained by an Autoflex II MALDI instrument (Bruker Daltonics, Bremen, Germany). 2,5-Dihydroxy-benzoic acid (DHB) was used for the ionization of the samples. Mass spectra were monitored either in positive or in negative mode (depending on the chemical structure) with pulsed ionization (λ = 337 nm; nitrogen laser). Spectra were measured in reflectron mode using a delayed extraction of 120 ns. The elemental analysis measurements were performed on a Fisons EA 1110 CHNS apparatus (Thermo Scientific, Waltham, MA, US).

Thin layer chromatography was performed on TLC Silica gel 60 F_254_ on Al sheets (Merck, Darmstadt, Germany), and the spots were visualized under UV light. Preparative layer chromatography was executed on PLC Silica gel 60 F_254_ 1 mm on glass plate (Merck). For column chromatography, a Kieselgel 60 (Merck, particle size 0.063–0.200 mm) was used. All reagents used for synthesis were commercial and of analytically pure quality and all organic solvents were of HPLC grade. Organic solutions were dried over anhydrous Na_2_SO_4_ and concentrated under reduced pressure at 40 °C (bath temperature). Carotenoid pentynoates were prepared by a published method [28].

### 4.2. Assay for the Antioxidant Activity of the Carotenoids/TEAC Assay

A literature procedure [31] was applied with modifications. The 2,2′-azino-bis(3-ethylbenzothiazoline-6-sulfonic acid) radical cation (ABTS^•+^) was produced in aqueous solution by reacting ABTS^•+^ in 7 mM and potassium persulfate in 2.45 mM concentrations. The stock ABTS^•+^ solution was prepared 12–16 h before the experiments, and stored at room temperature in dark. The absorbance of the ABTS^•+^ solution was set to 0.70 ± 0.05 at 734 nm by dilution with ethanol. Trolox was dissolved in ethanol, and the carotenoids and the carotenoid conjugates in tetrahydrofuran (THF). The antioxidants (in final concentrations of 1.25, 2.5, 3.75, and 5 μM) were incubated with ABTS^•+^ solution at 37 °C for 6 min, respectively. For the statistical analysis, the measurements of absorbance were carried out at each concentration in triplicate (n = 3), and each measurement was repeated 2 times. SD% was lower than 5% in each case. The percentage inhibition of absorbance at 734 nm was calculated as (*A*_0_ − *A*_antioxidant_)/*A*_0_, where A_0_ is the absorbance of the ABTS^•+^ solution and *A*_antioxidant_ is the absorbance measured after the addition of the antioxidant. The calculated values were plotted against the final concentration of the antioxidants. The slopes of the curves were compared with that for trolox, and the trolox equivalent antioxidant capacity (TEAC) value is the ratio of the slopes for the antioxidant and for trolox.

Statistical analysis of the TEAC values was performed by the ANOVA method (Microsoft Excel 2016) (Table 2). To examine the relationship among the TEAC values of the carotenoids, flavonoids, and those of their conjugates, two-tailed *t* tests were calculated. An F test was performed in each case to determine the type of the *t* test to be used. Less than 5% in the *t* test was regarded as a significant difference (Table 3). For the calculations, the functions of Microsoft Excel (2016) were used.

### 4.3. Acetylation of Flavonoids

Daidzein (**1**) or chrysin (**7**) (8 mmol) was dissolved in acetic anhydride (30 mL), and was stirred at 60 °C for 10 min. Dry pyridine (8 mL) was added dropwise to the stirred mixture at 60 °C. After 5 min, a white precipitate appeared. After 2 h, the reaction mixture was poured onto ice and stirred for an hour. The formed white precipitate was filtered out, and washed with water thoroughly. After drying above potassium hydroxide for 2 days, the crude products were recrystallized from ethanol.

*7-Acetoxy-3-(4-acetoxyphenyl)-4H-chromen-4-one* (**2**): white crystals (98%); mp 188–189 °C; UV (THF) *λ*_max_ 243, 277, 301 nm; IR (KBr) ν*_max_* 1752, 1617, 1646, 1224 cm^−1^; ^1^H-NMR (CDCl_3_, 500 MHz) *δ* 8.32 (1H, d, *J* = 8.7 Hz, H-5a), 8.00 (1H, s, H-2c), 7.58 (2H, d, *J* = 8.6 Hz, 2 H-2b), 7.31 (1H, d, *J* = 1.9 Hz, H-8a), 7.16–7.19 (3H, m, 2 H-3b, H-6a), 2.36 (3H, s, CH_3_), 2.32 (3H, s, CH_3_); ^13^C-NMR (CDCl_3_, 125 MHz) *δ* 175.4 (C, C-4c), 169.3, 168.4 (C, 2 Ac C=O), 156.7 (C, C-8c), 154.6 (C, C-7a), 153.1 (CH, C-2c), 150.8 (C, C-4b), 130.0 (CH, 2 C-2b), 129.2 (C, C-1b), 127.9 (CH, C-5a), 124.8 (C, C-3c), 122.3 (C, C-4a), 121.6 (CH, 2 C-3b), 119.6 (CH, C-6a), 110.9 (CH, C-8a), 21.1 (2 Ac CH_3_); MS (MALDI-TOF, positive mode, with DHB matrix) *m*/*z* 339.0 [M + H]^+^ 360.9 [M + Na]^+^ 376.9 [M + K]^+^; Anal. calcd. for C_19_H_14_O_6_: C 67.45, H 4.17, found C 67.05, H 4.03.

*5,7-Diacetoxy-2-phenyl-4H-chromen-4-one* (**8**): white crystals (95%); mp 195–196 °C; UV (THF) *λ*_max_ 243, 292 nm; IR (KBr) ν*_max_* 1769, 1630 cm^−1^; ^1^H-NMR (CDCl_3_, 500 MHz) *δ* 7.85 (2H, d, *J* = 8.0 Hz, 2 × H-2b), 7.55-7.49 (3H, m, 2 × H-3b, H-4b), 7.35 (1H, d, *J* = 2.1 Hz, H-6a), 6.85 (1H, d, *J* = 2.1 Hz, H-8a), 6.65 (1H, s, H-3c ), 2.44, 2.35 (2 × 3H, 2 s, 2 x CH_3_); ^13^C-NMR (CDCl_3_, 125 MHz) *δ* 176.3 (C, C-4c), 169.3, 167.9 (C, 2 × Ac C=O), 162.5 (C, C-2c), 157.7 (C, C-7a), 153.9 (C, C-8c), 150.2 (C, C-5a), 131.7 (CH, C-4b), 131.1 (C, C-1b), 129.1, 126.2 (CH, C-3b, C-2b), 115.0 (C, C-4a), 113.6 (CH, C-6a), 109.0 (CH, C-8a), 108.6 (CH, C-3c), 21.1, 21.0 (2 Ac CH_3_); MS (MALDI-TOF, positive mode, with DHB matrix) *m*/*z* 339.5 [M + H]^+^, 361.0 [M + Na]^+^ 377.0 [M + K]^+^; Anal. calcd. for C_19_H_14_O_6_: C 67.45, H 4.17, found C 67.23, H 4.16.

### 4.4. Selective Deacetylation of Flavonoid Diacetates

*7-Hydroxy-3-(4-acetoxyphenyl)-4H-chromen-4-one* (**3**): Daidzein diacetate (**2**) (0.412 g, 1.21 mmol) was dissolved in 20 mL CH_2_Cl_2_, imidazole (0.248 g, 3.6 mmol, 3 eq.) in 10 mL CH_2_Cl_2_ was added dropwise at −15 °C in ice/salt bath. The reaction mixture was stirred at room temperature for 2 h under N_2_. A white precipitate appeared. The reaction mixture was diluted with CH_2_Cl_2 _(100 mL), was washed with 3M HCl (3 × 100 mL) and water (100 mL), dried (Na_2_SO_4_), and the solvent was evaporated under reduced pressure. The crude products (0.41 g) were crystallized from EtOH giving white crystals (0.264 g, 73%); mp 231–232 °C; UV (THF) *λ*_max_ 243, 277(sh), 297, 307 nm; IR (KBr) ν*_max_* 3102, 1751, 1618, 1258 cm^−1^; ^1^H-NMR (DMSO-*d*_6_, 500 MHz) *δ* 10.82 (1H, s, OH), 8.41 (1H, s, H-2c), 8.00 (1H, d, *J* = 8.8 Hz, H-5a), 7.61 (2H, d, *J* = 8.6 Hz, H-3b), 7.18 (2H, d, *J* = 8.6 Hz, H-2b), 6.95 (1H, dd, *J* = 8.6, 2.2 Hz, H-6a), 6.89 (1H, d, *J* = 2.13 Hz, H-8a), 2.29 (3H, s, CH_3_); ^13^C NMR (DMSO-d_6_, 125 MHz) *δ* 174.2 (C, C-4c), 169.1 (C, Ac C=O), 162.5, 157.3 (C, C-8c, C-7a), 153.7 (CH, C-2c), 150.0 (C, C-4b), 129.9 (CH, 2 C-2b), 129.5 (C, C-1b), 127.2 (CH, C-5a), 122.7 (C, C-3c), 121.4 (CH, 2 C-3b), 116.5 (C, C-4a), 115.2 (CH, C-6a), 102.1 (CH, C-8a), 20.7 (CH_3_, Ac-CH_3_); MS (MALDI-TOF, positive mode, with DHB matrix) *m*/*z* 297.0 [M + H]^+^ 318.9 [M + Na]^+^ 336.9 [M + K]^+^; Anal. calcd. for C_17_H_12_O_5_: C 68.92, H 4.08, found C 68.90, H 4.05.

*5-Acetoxy-7-hydroxy-2-phenyl-4H-chromen-4-one* (**9**): Chrysin diacetate (**8**) (1.0 g, 2.96 mmol) was dissolved in 50 mL CH_2_Cl_2_, imidazole (0.603g, 8.87 mmol, 3 eq.) in 40 mL CH_2_Cl_2_ was added dropwise at −15 °C in ice/salt bath. The reaction mixture was stirred at room temperature for 2 hours under N_2_. A white precipitate appeared, which was filtered out and washed with cold CH_2_Cl_2_. The crude product (0.79 g) was recrystallized from EtOH and yielded white crystals (0.42 g, 48%). mp 185–186 °C; UV (THF) *λ*_max_ 208, 243, 301 nm; IR (KBr) ν*_max_* 1770, 1628, 1282 cm^−1^; ^1^H NMR (DMSO-*d_6_*, 500 MHz) *δ* 11.10 (1H, s, OH), 8.02 (2H, d, *J* = 6.5 Hz, 2 × H-2b), 7.59-7.53 (3H, m, 2 × H-3b, H-4b), 6.94 (1H, d, *J* = 2.3 Hz, H-6a*), 6.75 (1H, s, H-3c ), 6.57 (1H, d, *J* = 2.3 Hz, H-8a*), 2.30 (3H, s, CH_3_) *interchangeable; ^13^C NMR (DMSO-*d_6_*,, 125 MHz) *δ* 175.2 (C, C-4c), 168.7 (C, Ac C=O), 162.2 (C, C-2c), 160.9 (C, C-7a), 158.2 (C, C-8a), 150.0 (C, C-5a), 131.5 (CH, C-4b), 130.8 (C, C-1b), 129.0, 126.1 (CH, C-2b, C-3b), 109.4 (C, C-4a), 108.6 (CH, C-6a*), 107.4 (CH, C-3c), 100.8 (CH, C-8a*), 20.9 (CH_3_, Ac CH_3_) *interchangeable; MS (MALDI-TOF, positive mode, with DHB matrix) *m*/*z* 297.0 [M + H]^+^ 319.9 [M + Na]^+^ 336.9 [M + K]^+^; Anal. calcd. for C_17_H_12_N_3_O_5_: C 68.92, H 4.08, found C 68.80, H 3.86.

### 4.5. Mitsunobu Reactions

Monoacetates of daidzein (**3**) or chrysin (**9**) (6.76 mmol) and triphenylphosphine (7.44 mmol, 1.1 eq.) were suspended in freshly distilled THF (100 mL). Then 6-bromo-1-hexanol (0.87 mL, 6.76 mmol, 1 eq.) was added, and diisopropyl azodicarboxylate (1.44 mL, 7.44 mmol, 1.1 eq.) were dropped to the stirred solution. After the dissolution of the starting materials, the yellow solution was stirred overnight under nitrogen, at room temperature. The reaction mixture was evaporated under reduced pressure, and the residue was purified by column chromatography (DCM-MeOH, 150:1).

*7-(6-Bromohexyloxy)-3-(4-acetoxyphenyl)-4H-chromen-4-one* (**4**): white crystals (71%); mp 164–165 °C; UV (THF) *λ*_max_ 243, 277, 296, 306 nm; IR (KBr) ν*_max_* 1759, 1633, 1266 cm^−1^; ^1^H-NMR (CDCl_3_, 500 MHz) *δ* 8.20 (1H, d, *J* = 8.7 Hz, H-5a), 7.94 (1H, s, H-2c), 7.58 (2H, d, *J* = 7.6 Hz, 2 H-3b), 7.16 (2H, d, *J* = 7.2 Hz, 2 H-2b), 7.00 (1H, d, *J* = 8.8 Hz, H-6a), 6.84 (1H, s, H-8a), 4.07 (2H, t, *J* = 6.3 Hz, O-C*H*_2_), 3.44 (2H, t, *J* = 6.3 Hz, Br-C*H*_2_), 2.32 (3H, s, Ac CH_3_), 1.92–1.86 (4H, m, CH_2_-β,γ), 1.54–1.58 (4H, m, CH_2_-α,δ); ^13^C-NMR (CDCl_3_, 125 MHz) *δ* 175.4 (C, C-4c), 169.3 (C, Ac C=O), 163.6 (C, C-7a), 157.9 (C, C-8c), 152.6 (CH, C-2c), 150.6 (C, C-4b), 130.0 (CH, 2 C-3b), 129.6 (C, C-1b), 127.8 (CH, C-5a), 124.58 (C, C-3c), 121.6 (CH, 2 C-2b), 118.3 (C, C-4a), 114.9 (CH, C-6a), 100.7 (CH, C-8a), 68.5 (O-*C*H_2_), 33.6 (Br-*C*H_2_), 32.6, 28.8, 27.9, 25.2 (CH_2_, α,β,γ,δ-CH_2_), 21.1 (Ac *C*H_3_); MS (MALDI-TOF, positive mode, with DHB matrix) *m*/*z* 458.9 [M + H]^+^, 480.9 [M + K]^+^, 498.7 [M + K]^+^; Anal. calcd. for C_23_H_23_BrO_5_: C 60.14, H 5.05, found C 60.15, H 4.98.

*5-Acetoxy-7-(6-bromohexyloxy)-2-phenyl-4H-chromen-4-one* (**10**): white crystals (68%); mp 102–103 °C; UV (THF) *λ*_max_ 205, 216, 242, 300 nm; IR (KBr) ν*_max_* 1775, 1635, 1207 cm^−1^; ^1^H-NMR (CDCl_3_, 500 MHz) *δ* 7.85 (2H, d, *J* = 7.1 Hz, 2 x H-2b), 7.53-7.50 (3H, m, 2 × H-3b, H-4b), 6.86 (1H, s, H-3c*), 6.60 (2H, ps, H-6a*, H-8a*), 4.06 (2H, t, *J* = 6.2 Hz, O-CH_2_), 3.43 (2H, t, *J* = 6.6 Hz, Br-C*H*_2_), 2.44 (3H, s, Ac CH_3_), 1.92–1.84 (4H, m, 2 × CH_2_), 1.52 (4H, ps, 2 × CH_2_) *interchangeable; ^13^C-NMR (CDCl_3_, 125 MHz) *δ* 176.5 (C, C-4c), 169.6 (C, Ac C=O), 162.9 (C, C-2c), 161.9 (C, C-7a), 158.8 (C, C-8c), 150.5 (C, C-5a), 131.4 (CH, C-4b), 129.0, 126.1 (CH, C-2b, C-3b), 111.1 (C, C-4a), 108.6 (CH, C-6a*), 108.4 (CH, C-3c*), 99.4 (CH, C-8a), 68.6 (CH_2_, O-CH_2_), 33.6 (CH_2_, Br-CH_2_-), 32.6, 28.7, 27.8, 25.2 (CH_2_, 4 CH_2_), 21.1 (Ac CH_3_) *interchangeable; MS (MALDI-TOF, positive mode, without matrix) *m*/*z* 458.6 [M + H]^+^ 480.6 [M + Na]^+^ 498.6 [M + K]^+^; Anal. calcd. for C_23_H_23_BrO_5_: C 60.14, H 5.05, found C 60.44, H 4.94.

### 4.6. Deprotection

The acetylated 7-bromohexyl ether of daidzein (**4**) or chrysin (**10**) (5.0 mmol) was dissolved in dry MeOH (80 mL), and freshly prepared sodium methylate solution (10 mL) was added dropwise to the mixture, which was stirred at room temperature for 2 h. The pH of the mixture was controlled and the amount of NaOMe was replaced due to the formation of phenolates. After the complete disappearance of the starting material, the solution was neutralized by Amberlite IRC 86 acidic resin, and was evaporated in vacuum.

*7-(6-Bromohexyloxy)-3-(4-hydroxyphenyl)-4H-chromen-4-one* (**5**): white crystals (97%); mp 151–152 °C; UV (THF) *λ*_max_ 243, 277, 292, 306 nm; IR (KBr) ν*_max_* 3210, 1628, 1260 cm^−1^; ^1^H-NMR (CDCl_3_, 500 MHz) *δ* 8.21 (1H, d, *J* = 8.9 Hz, H-5a), 7.92 (1H, s, H-2c), 7.37 (2H, d, *J* = 8.6 Hz, 2 H-3b), 6.99 (1H, dd, *J* = 2.3, 8.9 Hz, H-6a), 6.82-6.85 (3H, m, H-8a, 2 H-2b), 6.04 (1H, s, OH), 4.07 (2H, t, *J* = 6.4 Hz, O-C*H*_2_), 3.44 (2H, t, *J* = 6.7 Hz, Br-C*H*_2_), 1.92 (2H, dt, *J* = 6.9, 13.9 Hz, CH_2_), 1.87 (2H, dt, *J* = 6.7, 13.1 Hz, CH_2_), 1.54 (4H, m, 2 CH_2_); ^13^C NMR (CDCl_3_, 125 MHz) *δ* 176.3 (C, C-4c), 163.6 (C, C-7a), 158.1 (C, C-8c), 156.0 (C, C-4b), 152.3 (CH, C-2c), 130.3 (CH, C-3b), 127.8 (CH, C-5a), 125.1 (C, C-1b), 123.9 (C, C-3c), 118.3 (C, C-4a), 115.7 (CH, C-2b), 115.0 (CH, C-6a), 100.6 (CH, C-8a), 68.5 (O-*C*H_2_), 33.7 (Br-*C*H_2_), 32.7, 28.8, 27.9, 25.3 (4 CH_2_); MS (MALDI-TOF, positive mode, with DHB matrix) *m*/*z* 417,3 [M + H]^+^, 438.6 [M + Na]^+^, 455.1 [M + K]^+^; Anal. calcd. for C_21_H_21_BrO_4_: C 60.44, H 5.07, found C 60.38, H 5.09.

*5-Hydroxy-7-(6-bromohexyloxy)-2-phenyl-4H-chromen-4-one* (**11**): white powder (96%); mp 129–130 °C; UV (THF) *λ*_max_ 243, 290, 308 nm; IR (KBr) ν*_max_* 3064, 1660, 1615, 1174 cm^−1^; ^1^H-NMR (CDCl_3_, 500 MHz) *δ* 12.70 (1H, s, OH), 7.88 (2H, dd, *J* = 1.3 Hz, *J* = 7.8 Hz, 2 × H-2b), 7.55–7.50 (3H, m, 2 × H-3b, H-4b), 6.66 (1H, s, H-3c), 6.49 (1H, d, *J* = 2.2 Hz, H-8a), 6.35 (1H, d, *J* = 2.2 Hz, H-6a), 4.04 (2H, t, *J* = 6.4 Hz, O-CH_2_), 3.44 (2H, t, *J* = 6.7 Hz, Br-C*H*_2_), 1.94–1.81 (4H, m,2 × CH_2_), 1.53–1.52 (4H, m, 2 × CH_2_); ^13^C-NMR (CDCl_3_, 125 MHz) *δ* 176.3 182.4 (C, C-4c), 165.1 (C, C-7a), 163.91 (C, C-2c), 162.2 (C, C-5a), 157.8 (C, C-8c), 131.8 (CH, C-4b), 131.4 (C, C-1b), 129.1 (CH, C-3b), 126.3 (CH, C-2b), 105.9 (CH, C-3c), 105.6 (C, C-4a), 98.6 (CH, C-6a), 93.1 (CH, C-8a), 68.4 (CH_2_, O-CH_2_), 33.7 (CH_2_, Br-CH_2_-), 32.6, 28.8, 27.8, 25.2 (CH_2_, 4 × CH_2_); MS (MALDI-TOF, positive mode, without matrix) *m*/*z* 417.9 [M + H]^+^, 456.4 [M + K]; Anal. calcd. for C_21_H_21_BrO_4_: C 60.44, H 5.07, found C 60.29, H 4.96.

### 4.7. Azide Substitutions

The 7-bromohexyl ether of daidzein (**5**) or chrysin (**11**) (4.0 mmol) and sodium azide (2.6 g, 40 mmol, 10 eq.) were dissolved in dry DMF. The reaction mixture was stirred at 60 °C for 2 days under N_2_. When the TLC showed the disappearance of the starting material, the excess sodium-azide and byproducts were filtered out. The mother liquor was diluted with CHCl_3_ (600 mL) and washed with water (3 × 200 mL). The organic phase was dried on Na_2_SO_4_ and filtered. The solvent was evaporated under reduced pressure, and the crude product was recrystallized from EtOH.

*7-(6-Azidohexyloxy)-3-(4-hydroxyphenyl)-4H-chromen-4-one* (**6**): white flakes (65%); mp 144–145 °C; UV (THF) *λ*_max_ 243, 277 (sh), 291 nm; IR (KBr) ν*_max_* 3217, 2104, 2150, 1628 cm^−1^; ^1^H NMR (DMSO-*d*_6_, 500 MHz) *δ* 9.53 (1H, s, OH), 8.34 (1H, s, H-2c), 8.00 (1H, d, *J* = 8.8 Hz, H-5a), 7.40 (2H, d, *J* = 8.6 Hz, H-3b), 7.11 (1H, d, *J* = 2.3 Hz, H-8a), 7.05 (1H, dd, *J* = 2.3, 8.9 Hz, H-6a), 6.81 (2H, d, *J* = 8.6 Hz, H-2b), 4.10 (2H, t, *J* = 6.4 Hz, O-C*H*_2_), 3.33 (2H, t, *J* = 6.9 Hz, N_3_-C*H*_2_), 1.76 (2H, pddd, *J* = 6.6, 13.4 Hz, α-CH_2_), 1.56 (2H, pddd, *J* = 7.0, 14.2 Hz, δ-CH_2_), 1.36-1.48 (4H, m, β,γ-CH_2_); ^13^C-NMR (DMSO-*d*_6_, 125 MHz) *δ* 174.5 (C, C-4c), 162.9 (C, C-7a), 157.2 (C, C-8c), 157.1 (C, C-4b), 152.8 (CH, C-2c), 129.9 (CH, C-3b), 126.7 (CH, C-5a), 123.5 (C, C-1b), 122.2 (C, C-3c), 117.4 (C, C-4a), 114.8 (CH, C-2b), 114.7 (CH, C-6a), 100.8 (CH, C-8a), 68.2 (O-*C*H_2_), 50.5 (N_3_-*C*H_2_), 28.1 (α-CH_2_), 28.0 (δ-CH_2_), 25.7(γ-CH_2_), 24.8 (β-CH_2_); MS (MALDI-TOF, positive mode, with DHB matrix) *m*/*z* 379.7 [M + H]^+^, 417.5 [M + K]^+^; Anal. calcd. for C_12_H_21_N_3_O_4_: C 66.48, H 5.58, N 11.08 found C 66.23, H 5.53 N 10.82.

*5-Hydroxy-7-(6-azidohexyloxy)-2-phenyl-4H-chromen-4-one* (**12**): white flakes (52%); mp 97–98 °C; UV (THF) *λ*_max_ 243, 291, 309(sh) nm; IR (KBr) ν*_max_* 3060, 2131, 2085, 1653, 1603 cm^−1^; ^1^H-NMR (CDCl_3_, 500 MHz) *δ* 12.70 (1H, s, OH), 7.87 (2H, d, *J* = 7.0 Hz, 2 × H-2b), 7.56-7.50 (3H, m, 2 × H-3b, H-4b), 6.65 (1H, s, H-3c), 6.48 (1H, d, *J* = 1.8 Hz, H-8a), 6.35 (1H, d, *J* = 1.8 Hz, H-6a), 4.03 (2H, t, *J* = 6.4 Hz, O-CH_2_), 3.30 (2H, t, *J* = 6.8 Hz, N_3_-CH_2_), 1.86–1.80 (2H, m, CH_2_), 1.68–1.62 (2H, m, CH_2_), 1.55–1.45 (4H, m, 2 CH_2_); ^13^C NMR (CDCl_3_, 125 MHz) *δ* 182.40 (C, C-4c), 165.07 (C, C-7a), 163.89 (C, C-2c), 162.17 (C, C-5a), 157.78 (C, C-8c), 131.75 (CH, C-4b), 131.36 (C, C-1b), 129.04 (CH, C-3b), 126.24 (CH, C-2b), 105.85 (CH, C-3c), 105.63 (C, C-4a), 98.56 (CH, C-6a), 93.07 (CH, C-8a), 68.39 (CH_2_, O-CH_2_-), 51.35 (CH_2_, N_3_-CH_2_-), 28.82, 28.77, 26.44, 25.57 (CH_2_, 4 CH_2_); MS (MALDI-TOF, positive mode, with DHB matrix) *m*/*z* 380.1 [M + H]^+^, 417.9 [M + K]^+^; Anal. calcd. for C_21_H_21_N_3_O_4_: C 66.48, H 5.58, N 11.08 found C 66.57, H 5.06 N 10.99.

### 4.8. Azide-Alkyne Click-Reactions

The carotenoid pentynoates (**13**,**14**,**15**) (0.1 mmol) were dissolved in dry dichloromethane (4 mL), and the 7-azidohexyl ether derivatives of flavonoids (**6**,**12**) (1.5 or 2.5 eq.) and bistriphenylphosphano-copper(I)-butyrate (C_3_H_7_COOCu(PPh_3_)_2_) (6 mg) were added. The reaction mixtures were stirred under nitrogen, in darkness. When TLC showed the disappearance of the starting materials (typically in 24 h), the solvent was evaporated in vacuum, and the residue was purified by column chromatography. After crystallization, the products were stored in closed ampoules under argon. Yields are given for the bistriazoles and monotriazoles separately. The analytical samples were further purified by preparative TLC.

*8′-(1-(6-(3-(4-hydroxyphenyl)-4H-chromen-4-one-7-yloxy)hex-1-yl)-1,2,3-triazol-4-yl)-propanoyloxy)-8′-apo-β-carotene* (**16**): red crystals (62%); mp 118–119 °C; UV (THF) *λ*_max_ 242, 278, 289, 375, 402, 429, 455 nm; IR (KBr) ν*_max_* 3414, 1734, 1626 cm^−1^; ^1^H-NMR (CDCl_3_, 500 MHz) *δ* 8.18 (1H, dd, *J* = 1.3, 8.9 Hz, H-5a), 7.88 (1H, d, *J* = 1.3 Hz, H-2c), 7.37–7.39 (2H, m, H-3b), 7.34 (1H, s, triazole CH), 6.95 (1H, d, *J* = 8.9 Hz, H-6a), 6.87 (2H, d, *J* = 8.6 Hz, H-2b), 6.80 (1H, s, H-8a), 6.68–6.57 (3H, m, H-11, H-15, H-15′), 6.43 (1H, dd, *J* = 11.0, 15.0 Hz, H-11′), 6.34 (1H, d, *J* = 9.9 Hz, H-12), 6.31 (1H, d, *J* = 10.1 Hz, H-12′), 6.25-6.22 (2H, m, H-14, H-14′), 6.20–6.11 (4H, m, H-7, H-8, H-10, H-10’), 4.56 (2H, s, H-8′), 4.32 (2H, t, *J* = 7.1 Hz, N-CH_2_), 4.01 (2H, t, *J* = 4.0 Hz, O-CH_2_), 3.06 (2H, t, *J* = 7.2 Hz, C*H*_2_-C(=C)N), 2.78 (2H, t, *J* = 7.2 Hz, C*H*_2_-C=O), 2.02 (2H, t, *J* = 6.0 Hz, 2 H-4), 1.97 (6H, s, 2 CH_3_), 1.93 (3H, s, CH_3_), 1.80 (3H, s, CH_3_), 1.71 (3H, s, CH_3_), 1.64-1.60 (2H, m, 2 H-3), 1.55-1.45 (4H, m, CH_2_-β, 2 H-2), 1.42-1.36 (2H, m, CH_2_-γ), 1.26 (2H, ps, CH_2_-α), 1.03 (6H, s, 2 CH_3_); ^13^C-NMR (CDCl_3_, 125 MHz) *δ* 176.1 (C, C-4c), 172.6 (C, O*C*=O), 163.4 (C, C-7a), 158.0 (C, C-4b), 156.4 (C, C-8c), 152.1 (CH, C-2c), 146.3 (C, triazole N*C*=C), 138.5 (CH, C-12′), 137.9 (C, C-6), 137.7 (CH, C-8), 137.1 (CH, C-12), 136.8 (C, C-13), 136.2, 135.7 (C, C-9, C-13′ interchangeable), 133.0, 132.2 (CH, C-14, C-14′ interchangeable), 131.7 (C, C-9′), 130.7 (CH, C-10), 130.5, 129.6 (CH, C-15, C-15′ interchangeable), 130.2 (CH, C-3b), 129.4 (C, C-5), 129.0 (CH, C-10’), 127.7 (CH, C-5a), 126.8 (CH, C-7), 125.3 (CH, C-11), 125.0 (C, C-1b), 123.7 (C, C-3c), 123.3 (CH, C-11′), 121.2 (CH, triazole), 118.3 (C, C-4a), 115.7 (CH, C-2b), 114.8 (CH, C-6a), 100.6 (CH, C-8a), 69.9 (CH_2_, C-8′), 68.3 (*C*H_2_-O), 50.1 (CH_2_-N), 39.7 (CH_2_, C-2), 34.3 (C, C-1), 33.7 (*C*H_2_-CO), 33.1 (CH_2_, C-4), 30.1 (CH_2_, CH_2_-δ), 29.0 (CH_3_, C-16, C-17), 28.8 (CH_2_, CH_2_-α), 26.2 (CH_2_, CH_2_-γ), 25.4 (CH_2_, CH_2_-β), 21.7 (CH_3_, CH_3_-18), 21.0 (CH_2_, *C*H_2_-C(=C)N), 19.3 (CH_2_, C-3), 14.7 (CH_3_, CH_3_-19′), 12.81, 12.75 (CH_3_, CH_3_-19,20,20’); MS (MALDI-TOF, positive mode, with DHB matrix) *m*/*z* 878.7 [M + H]^+^; Anal. calcd. for C_56_H_67_N_3_O_6_: C 76.59, H 7.69, N 4.79 found C 76.12, H 7.47 N 4.23.

*3,3′-bis(1-(6-(3-(4-hydroxyphenyl)-4H-chromen-4-one-7-yloxy)hex-1-yl)-1,2,3-triazol-4-yl)-propanoyloxy)-β,β-carotene* (**17**): red crystals (42%); mp 177–178 °C; UV (THF) *λ*_max_ 243, 278, 429(sh), 456, 486 nm; IR (KBr) ν*_max_* 3414, 2919, 1730, 1624, 1253 cm^−1^; ^1^H-NMR (CDCl_3_, 500 MHz) *δ* 8.16 (2H, dd, *J* = 5.3, 8.9 Hz, 2 × H-5a), 7.88 (2H, d, *J* = 3.1 Hz, 2 x H-2c), 7.37 (4H, d, *J* = 3.8 Hz, 4 × H-3b), 7.36 (2H, s, 2 × triazole CH), 6.94 (2H, dd, *J* = 1.9 Hz, *J* = 8.8 Hz, 2 × H-6a), 6.87 (4H, d, *J* = 7.9 Hz, 4 × H-2b), 6.80 (2H, s, 2 × H-8a), 6.67–6.59 (4H, m, H-11,11′, H-15,15′), 6.34 (2H, d, *J* = 14.8 Hz, H-12,12′), 6.27–6.21 (2H, m, H-14,14′), 6.14 (2H, d, *J* = 11.4 Hz, H-10,10’), 6.11–6.05 (4H, m, H-7,7′, H-8,8′), 5.09-5.03 (2H, m, H-3,3′), 4.34 (4H, t, *J* = 7.0 Hz, 2 × N-CH_2_), 4.01 (4H, t, *J* = 5.5 Hz, 2 × O-CH_2_), 3.05 (4H, t, *J* = 7.1 Hz, 2 × C*H*_2_-C(=C)N), 2.73 (4H, t, *J* = 7.0 Hz, 2 × C*H*_2_-C=O), 2.39 (2H, dd, *J* = 5.3 Hz, *J* = 16.9 Hz, H-4_eq_, H-4′_eq_), 2.11–2.04 (2H, m, H-4_ax_, H-4′_ax_), 1.95 (12H, s, CH_3_-19,19′, CH_3_-20,20’,), 1.82–1.74 (10H, m, H-2_eq_, H-2′_eq_, 2 × 2 CH_2_), 1.70 (6H, s, CH_3_-18,18′), 1.58–1.48 (6H, m, H-2_ax_, H-2′_ax_, 2 × CH_2_), 1.42–1.35 (4H, m, 2 × CH_2_), 1.09, 1.05 (12H. 2 s, CH_3_-16,16′, CH_3_-17,17′); ^13^C-NMR (CDCl_3_, 125 MHz) *δ* 176.2 (C, C-4c), 172.2 (C, O*C*=O), 163.4 (C, C-7a), 158.0, 156.6 (C, C-4b, C-8c interchangeable), 152.2 (CH, C-2c), 146.5 (C, triazole N*C*=C), 138.71, 137.7 (CH, C-12,12′ and C-8,8′), 137.9, 136.5, 135.5, 125.5 (C, C-5,5′, C-6,6′, C-9,9′, C-13,13′ interchangeable), 132.7 (CH, C-14,14′), 131.5, 130.1 (CH, C-10,10’ and C-15,15′ interchangeable), 130.2 (CH, C-3b), 127.7 (CH, C-5a), 125.2, 124.9 (CH, C-7,7′ and C-11,11′ interchangeable), 125.1 (C, C-1b), 123.4 (C, C-3c), 121.2 (CH, triazole), 118.2 (C, C-4a), 115.7 (CH, C-2b), 114.9 (CH, C-6a), 100.6 (CH, C-8a), 68.7 (CH, C-3,3′), 68.3 (CH_2_-O), 50.1 (CH_2_-N), 44.0 (CH_2_, C-2,2′), 38.4 (CH_2_, C-4,4′), 36.7 (C, C-1,1′), 34.0 (*C*H_2_-CO), 30.1, 28.8, 26.2, 25.4 (CH_2_, CH_2_-α,β,γ,δ interchangeable), 30.0, 28.5 (CH_3_, CH_3_-16,16′ and CH_3_-17,17′ interchangeable), 21.5 (CH_3_, CH_3_-18,18′), 21.0 (CH_2_, *C*H_2_-C(=C)N), 12.8, 12.7 (CH_3_, CH_3_-19,19′ and CH_3_-20,20’ interchangeable); MS (MALDI-TOF, positive mode, with DHB matrix) *m*/*z* 1509.794 [M + Na]^+^; Anal. calcd. for C_92_H_106_N_6_O_12_: C 74.27, H 7.18, N 5.65 found C 74.13, H 7.79 N 5.56.

*3-(1-(6-(3-(4-hydroxyphenyl)-4H-chromen-4-one-7-yloxy)hex-1-yl)-1,2,3-triazol-4-yl)-propanoyloxy)-3′-(4-pentynoyloxy)-β,β-carotene* (**18**): red crystals (7%); mp 155–156 °C; UV (THF) *λ*_max_ 242, 279, 340, 428(sh), 455, 484 nm; IR (KBr) ν*_max_* 3420, 1730, 1624, 1253 cm^−1^; ^1^H-NMR (CDCl_3_, 500 MHz) *δ* 8.19 (1H, d, *J* = 8.9 Hz, H-5a), 7.90 (1H, s, H-2c), 7.42 (2H, d, *J* = 8.5 Hz, H-3b), 7.36 (1H, s, triazole CH), 6.96 (1H, dd, *J* = 2.2, 8.9 Hz, H-6a), 6.88 (2H, d, *J* = 8.5 Hz, H-2b), 6.81 (1H, d, *J* = 2.1 Hz, H-8a), 6.70–6.60 (4H, m, H-11,11′, H-15,15′), 6.36 (2H, dd, *J* = 2.8, 14.7 Hz, H-12,12′), 6.25 (2H, d, *J* = 7.5 Hz, H-14,14′), 6.16 (2H, dd, *J* = 4.0, 11.6 Hz, H-10,10’), 6.12–6.09 (4H, m, H-7,7′ and H-8,8′), 5.13–5.03 (2H, m, H-3,3′), 4.34 (2H, t, *J* = 7.1 Hz, O-CH_2_), 4.03 (2H, t, *J* = 6.2 Hz, N-CH_2_), 3.06 (2H, t, *J* = 7.3 Hz, C*H*_2_-C(=C)N), 2.73 (2H, t, *J* = 7.2 Hz, C*H*_2_-C=O), 2.56–2.46 (4H, m, 2 CH_2_ pentynoate), 2.47–2.38 (2H, m, H-4,4′_eq_), 2.17–2.07 (2H, m, H-4,4′_ax_), 1.98 (1H, s, C≡CH), 1.97 (12H, ps, 4 CH_3_-19,19′,20,20’), 1.86–1.76 (6H, m, H-2,2′_eq_, a linker CH_2_), 1.72, 1.71 (6H,2 s, CH_3_-18,18′), 1.56–1.51 (2H, m, H-2,2′_ax_), 1.42–1.38 (4H, m, 2 linker CH_2_), 1.27–1.20 (2H, m, a linker CH_2_), 1.11, 1.10, 1.08, 1.06 (12H, 4 s, CH_3_-16,16′,17,17′); MS (MALDI-TOF, positive mode, with DHB matrix) *m*/*z* 1130.7 [M + Na]^+^; Anal. calcd. for C_71_H_85_N_3_O_8_: C 76.93, H 7.73, N 3.79 found C 76.18, H 7.35 N 3.62.

*3,3′-bis(1-(6-(3-(4-hydroxyphenyl)-4H-chromen-4-one-7-yloxy)hex-1-yl)-1,2,3-triazol-4-yl)-propanoyloxy)-β,κ-carotene* (**19**): dark red crystals (40%); mp 86 °C decomp.; UV (THF) *λ*_max_ 242, 288, 360, 475 nm; IR (KBr) ν*_max_* 3414, 1734, 1626, 1252 cm^−1^; ^1^H-NMR (CDCl_3_, 500 MHz) *δ* 8.17 (2H, dd, *J* = 2.5, 8.9 Hz, 2 × H-5a), 7.88 (2H, d, *J* = 2.2 Hz, 2 × H-2c), 7.37–7.35 (7H, m, 4 × H-3b, H-8′, 2 × triazole CH), 6.94 (2H, d, *J* = 8.9 Hz, 2 × H-6a), 6.89–6.87 (4H, m, 2 × H-2b), 6.79 (2H, d, *J* = 2.1 Hz, 2 × H-8a), 6.70–6.47 (6H, m, H-10’,11,11′,12′,15,15′), 6.42–6.32 (3H, m, H-7′,12,14′), 6.24 (1H, d, *J* = 11.3 Hz, H-14), 6.14 (1H, d, J = 11.3 Hz, H-10), 6.12-6.09 (2H, m, H-7, H-8), 5.26-5.23 (1H, m, H-3′), 5.09-5.04 (1H, m, H-3), 4.33 (4H, dd, *J* = 7.0 Hz, *J* = 13.7.0 Hz, 2 × N-CH_2_), 4.00 (4H, t, *J* = 4.6 Hz, 2 × O-CH_2_), 3.04 (4H, dd, *J* = 6.8 Hz, *J* = 12.9 Hz, C*H*_2_-C(=C)N), 2.95 (1H, dd, *J* = 8.8 Hz, *J* = 14.9 Hz, H-4′_α_), 2.74–2.69 (4H, m, 2 × C*H*_2_-C=O), 2.39 (1H, dd, *J* = 5.3 Hz, *J* = 16.8 Hz, H-4_eq_), 2.11–2.02 (2H, m, H-2′_α_, H-4_ax_), 1.96, 1.95, 1.94, 1.93 (12H, 4s, CH_3_-19,19′,20,20’), 1.82–1.74 (6H, m, H-2_eq_, H-2′_β_, 2 CH_2_), 1.70 (3H, s, CH_3_-18), 1.58–1.47 (6H, m, H-2_ax_, H-4′_β_, 2 CH_2_), 1.42–1.36 (4H, m, 2 CH_2_), 1.28 (3H, s, CH_3_-18′), 1.26–1.22 (4H, m, 2 CH_2_), 1.15, 1.09 (6H, 2 s, CH_3_-16′,17′), 1.06, 0.84 (6H, 2 s, CH_3_-16,17); ^13^C NMR (CDCl_3_, 125 MHz) *δ* 202.5 (C, C-6′), 176.2 (C, 2 × C-4c), 172.5 (C, 2 × O*C*=O), 163.4 (C, 2 × C-7a), 158.0 (C, 2 × C-4b), 156.5 (C, 2 × C-8c), 152.2 (CH, 2 x C-2c), 147.2 (CH, C-8′), 146.5, 146.42 (C, 2 triazoles N*C*=C), 142.1, 141.0, 138.6, 137.5, 135.4, 132.5, 131.7, 131.4, 129.8, 125.5, 124.0, 121.3, 121.2 (CH, 2 × triazoles *C*H, C-7,8,10,10’,11,11′,12,12′,14,14′,15,15′), 137.9, 137.6, 136.0, 135.9, 133.5, 125.6, 125.1, 123.6, (C, 2 × C-3c, 2 C-1b, C-5,6,9,9′,13,13′), 130.2 (CH, 4 × C-3b), 127.73, 127.71 (CH, 2 × C-5a), 120.6 (CH, C-7′), 118.2 (C, 2 × C-4a), 115.7 (CH, 4 × C-2b), 114.9, 114.8 (CH, 2 × C-6a), 100.6 (CH, C-8a), 73.8 (CH, C-3′), 68.7 (CH, C-3), 68.4 (CH_2_, O-CH_2_), 58.6 (C, C-5′), 50.1 (CH_2_, N-CH_2_), 47.6 (CH_2_, C-2′), 44.0 (CH_2_, C-2), 43.7 (CH_2_, C-4′), 42.3 (CH_2_, C-4), 38.4, 36.7 (C, C-1,1′), 34.0, 33.9 (CH_2_, 2 × *C*H_2_-CO), 30.0, 28.6, 25.6, 24.8 (CH_3_, CH_3_-16,16′,17,17′), 30.2, 28.8, 26.2, 25.5 (CH_2_, 8 × CH_2_ linker), 21.5, 20.7 (CH_3_, CH_3_-18,18′), 21.1 (CH_2_, 2 × *C*H_2_-C=C), 12.9, 12.81, 12.80, 12.7 (CH_3_, CH_3_-19,19′,20,20’); MS (MALDI-TOF, positive mode, with DHB matrix) *m*/*z* 1525.8 [M + Na]^+^; Anal. calcd. for C_92_H_106_N_6_O_13_: C 73.48, H 7.10, N 5.59 found C 72.92, H 6.68 N 5.49.

*3-(1-(6-(3-(4-hydroxyphenyl)-4H-chromen-4-one-7-yloxy)hex-1-yl)-1,2,3-triazol-4-yl)-propanoyloxy)-3′-(4-pentynoyloxy)-β,κ-carotene* (**20a**): dark red crystals (22%); mp 86–87 °C decomp.; UV (THF) *λ*_max_ 242, 288, 476 nm; IR (KBr) ν*_max_* 3100–3400 (br), 3295, 2922, 1729, 1663, 1629, 1253 cm^−1^; ^1^H-NMR (CDCl_3_, 500 MHz) *δ* 8.19 (1H, d, *J* = 8.9 Hz, H-5a), 7.90 (1H, s, H-2c), 7.44 (2H, d, *J* = 8.6 Hz, H-3b), 7.36 (1H, s, triazole CH), 7.34 (1H, d, *J* = 14.8 Hz, H-8′), 6.96 (1H, dd, *J* = 2.1 Hz, *J* = 8.9 Hz, H-6a), 6.89 (2H, d, *J* = 8.6 Hz, H-2b), 6.81 (1H, d, *J* = 2.2 Hz, H-8a), 6.72–6.51 (6H, m, H-10’,11,11′,12′,15,15′), 6.43 (1H, d, *J* = 15.1 Hz, H-7′), 6.38–6.35 (2H, m, H-12,14′), 6.26 (1H, d, *J* = 11.2 Hz, H-14), 6.15 (1H, d, *J* = 12.5 Hz, H-10), 6.13–6.10 (2H, m, H-7, H-8), 5.31–5.25 (1H, m, H-3′), 5.13–5.03 (1H, m, H-3), 4.34 (2H, t, *J* = 7.1 Hz, N-CH_2_), 4.04 (2H, t, *J* = 6.3 Hz, O-CH_2_), 3.05 (4H, t, *J* = 7.3 Hz, C*H*_2_-C(=C)N), 2.99 (1H, dd, *J* = 8.4 Hz, *J* = 14.7 Hz, H-4′_α_), 2.73 (2H, t, *J* = 7.4 Hz, C*H*_2_-C=O), 2.52–2.50 (4H, m, CH_2_-pentynoate), 2.40 (1H, dd, *J* = 5.8 Hz, *J* = 17.9 Hz, H-4_eq_), 2.12–2.05 (2H, m, H-2′_α_, H-4_ax_), 1.98, 1.972, 1.965, 1.95 (13H, 4s, CH≡C, CH_3_-19,19′,20,20’), 1.85–1.73 (4H, m, H-2_eq_, H-2′_β_, CH_2_), 1.71 (3H, s, CH_3_-18), 1.69–1.51 (m, 2 × CH_2_ H-2_ax_, H-4′_β_, overlaid with the signal of water), 1.45–1.37 (4H, m, 2 CH_2_), 1.28 (3H, s, CH_3_-18′), 1.18, 1.10 (6H, 2 s, CH_3_-16′,17′), 1.06, 0.86 (6H, 2 s, CH_3_-16,17); MS (MALDI-TOF, positive mode, with DHB matrix) *m*/*z* 1146.7 [M + Na]^+^; Anal. calcd. for C_71_H_85_N_3_O_9_: C 75.84, H 7.62, N 3.74 found C 75.23, H 7.36 N 3.40.

*3-(4-pentynoyloxy)-3′-(1-(6-(3-(4-hydroxyphenyl)-4H-chromen-4-one-7-yloxy)hex-1-yl)-1,2,3-triazol-4-yl)-propanoyloxy)- β,κ-carotene* (**20b**): dark red crystals (23%); mp 130–131 °C decomp.; UV (THF) *λ*_max_ 205, 242, 288, 476 nm; IR (KBr) ν*_max_* 3100–3400 (br), 3305, 2921, 1727, 1663, 1623, 1253 cm^−1^; ^1^H-NMR (CDCl_3_, 500 MHz) *δ* 8.19 (1H, d, *J* = 9.0 Hz, H-5a), 7.90 (1H, s, H-2c), 7.44 (2H, d, *J* = 8.6 Hz, H-3b), 7.36 (1H, s, triazole CH), 7.33 (1H, d, *J* = 16.4 Hz, H-8′), 6.96 (1H, dd, *J* = 2.2 Hz, *J* = 8.9 Hz, H-6a), 6.89 (2H, d, *J* = 8.5 Hz, H-2b), 6.82 (1H, d, *J* = 2.1 Hz, H-8a), 6.72–6.50 (6H, m, H-10’,11,11′,12′,15,15′), 6.41 (1H, d, *J* = 15.1 Hz, H-7′), 6.38–6.34 (2H, m, H-12,14′), 6.26 (1H, d, *J* = 11.7 Hz, H-14), 6.16 (1H, d, *J* = 11.3 Hz, H-10), 6.12–6.08 (2H, m, H-7, H-8), 5.27–5.22 (1H, m, H-3′), 5.14–5.07 (1H, m, H-3), 4.33 (2H, t, *J* = 7.1 Hz, N-CH_2_), 4.04 (2H, t, *J* = 6.2 Hz, O-CH_2_), 3.03 (4H, t, *J* = 7.1 Hz, C*H*_2_-C(=C)N), 2.96 (1H, dd, *J* = 8.5 Hz, *J* = 14.5 Hz, H-4′_α_), 2.71 (2H, t, *J* = 7.2 Hz, C*H*_2_-C=O), 2.60–2.50 (4H, m, CH_2_-pentynoate), 2.45 (1H, dd, *J* = 6.4 Hz, *J* = 17.7 Hz, H-4_eq_), 2.07–2.03 (2H, m, H-2′_α_, H-4_ax_), 1.99, 1.97, 1.96, 1.95, 1.94 (13H, 5s, CH≡C, CH_3_-19,19′,20,20’), 1.86–1.76 (4H, m, H-2_eq_, H-2′_β_, CH_2_), 1.72 (3H, s, CH_3_-18), 1.69–1.65 (2H, m, CH_2_), 1.61–1.51 (4H, m, H-2_ax_, H-4′β, CH_2_ overlayed with the signal of water), 1.43–1.33 (4H, m, 2 CH_2_), 1.29 (3H, s, CH_3_-18′), 1.15, 1.11 (6H, 2 s, CH_3_-16′,17′), 1.08, 0.84 (6H, 2 s, CH_3_-16,17); MS (MALDI-TOF, positive mode, with DHB matrix) *m*/*z* 1146.8 [M + Na]^+^; Anal. calcd. for C_71_H_85_N_3_O_9_: C 75.84, H 7.62, N 3.74 found C 75.32, H 7.75 N 3.63.

*8′-(1-(6-((5-hydroxy-2-phenyl-4H-chromen-4-one-7-yl)oxy)hex-1-yl)-1,2,3-triazol-4-yl)-propanoyloxy)-8′-apo-β-carotene* (**21**): red crystals (75%); mp 113–114 °C; UV (THF) *λ*_max_ 242, 286, 407, 432, 456 nm; IR (KBr) ν*_max_* 3341, 2923, 1736, 1614 cm^−1^; ^1^H NMR (CDCl_3_, 500 MHz) *δ* 12.71 (1H, s, OH), 7.88 (2H, d, *J* = 6.7 Hz, 2 × H-2b), 7.56–7.50 (3H, m, 2 × H-3b, H-4b), 7.33 (1H, s, triazole CH), 6.66 (1H, s, H-3c), 6.67–6.56 (3H, m, H-11, H-15, H-15′), 6.48 (1H, d, *J* = 2.0 Hz, H-8a), 6.45 (1H, dd, *J* = 11.0, 15.0 Hz, H-11′), 6.35 (1H, d, *J* = 1.8 Hz, H-6a), 6.34–6.30 (2H, m, H-12,12′), 6.25–6.20 (2H, m, H-14, H-14′), 6.16–6.11 (4H, m, H-7, H-8, H-10, H-10’), 4.56 (2H, s, H-8′), 4.32 (2H, t, *J* = 7.1 Hz, N-CH_2_), 4.01 (2H, t, *J* = 6.3 Hz, O-CH_2_), 3.06 (2H, t, *J* = 7.1 Hz, C*H*_2_-C(=C)N), 2.78 (2H, t, *J* = 7.2 Hz, C*H*_2_-C=O), 2.06–2.01 (2H, m, 2 H-4), 1.96, 1.94 (6H+3H, 2 s, CH_3_-19,20,20’), 1.91 (2H, m, CH_2_-δ, overlayed with s at 1.94 ppm), 1.80 (3H, s, CH_3_-19′), 1.72 (3H, s, CH_3_-18), 1.64-1.59 (2H, m, 2 H-3), 1.54–1.50 (2H, m, CH_2_-β), 1.48-1.46 (2H, m, 2 H-2), 1.38–1.23 (4H, m, CH_2_-α, CH_2_-γ), 1.03 (6H, s, CH_3_-16,17); ^13^C NMR (CDCl_3_, 125 MHz) *δ* 182.4 (C, C-4c), 172.6 (C, C=O), 165.0 (C, C-7a), 164.0 (C, C-2c), 162.2 (C, C-5a), 157.8 (C, C-8c), 146.4 (C, triazole N*C*=C), 138.5 (CH, C-12′), 137.9 (C, C-6), 137.7 (CH, C-8), 137.1 (CH, C-12), 136.8 (C, C-13), 136.2 (C, C-9), 135.7 (C, C-13′), 133.0, 132.2 (CH, C-14,14′), 131.8 (CH, C-4b), 131.7 (C, C-9′), 131.4 (C, C-1b), 130.8 (CH, C-10), 130.5, 129.6 (CH, C-15,15′), 129.4 (C, C-5), 129.1 (CH, C-3b), 129.0 (CH, C-10’), 126.8 (CH, C-7), 126.3 (CH, C-2b), 125.3 (CH, C-11), 123.3 (CH, C-11′), 121.1 (CH, triazole), 105.9 (CH, C-3c), 105.7 (C, C-4a), 98.6 (CH, C-6a), 93.1 (CH, C-8a), 69.9 (CH_2_, C-8′), 68.3 (CH_2_, O-CH_2_), 50.0 (CH_2_, N-CH_2_), 39.7 (CH_2_, C-2), 34.3 (C, C-1), 33.7 (*C*H_2_-CO), 33.1 (CH_2_, C-4), 31.9 (CH_2_, CH_2_-δ), 29.0 (CH_3_, CH_3_-16,17), 28.8 (CH_2_, CH_2_-α), 26.2 (CH_2_, CH_2_-γ), 25.5 (CH_2_, CH_2_-β ), 21.7 (CH_3_, CH_3_-18), 21.1 (CH_2_, *C*H_2_-C(=C)N), 19.3 (CH_2_, C-3), 14.1 (CH_3_, C-19′), 12.81, 12.76 (CH_3_, CH_3_-19,20,20’); MS (MALDI-TOF, positive mode, with DHB matrix) *m*/*z* 916.441 [M + K]^+^; Anal. calcd. for C_56_H_67_N_3_O_6_: C 76.59, H 7.69, N 4.79 found C 76.21, H 7.41, N 4.37.

*3,3′-bis(1-(6-((5-hydroxy-2-phenyl-4H-chromen-4-one-7-yl)oxy)hex-1-yl)-1,2,3-triazol-4-yl)-propanoyloxy)-β,β-carotene* (**22**): orange-red crystals (59%); mp 92–93 °C; UV (THF) *λ*_max_ 243, 277, 432, 458, 486 nm; IR (KBr) ν*_max_* 2922, 1729, 1614 cm^−1^; ^1^H NMR (CDCl_3_, 500 MHz) *δ* 12.71 (2H, s, 2 OH), 7.87 (4H, d, *J* = 6.6 Hz, 2 × H-2b), 7.53–7.50 (6H, m, 4 × H-3b, 2 × H-4b), 7.35 (2H, s, 2 triazole CH), 6.65 (2H, s, H-3c), 6.63–6.61 (4H, m, H-11,11′,15,15′), 6.47 (2H, s, 2 × H-8a), 6.35 (4H, pd, *J* = 12.4 Hz, H-12,12′, 2 × H-6a), 6.24 (2H, d, *J* = 5.5 Hz, H-14,14′), 6.14 (2H, d, *J* = 11.1 Hz, H-10,10’), 6.10–6.05 (4H, m, H-7,7′,8,8′), 5.06 (4H, ps, H-3,3′), 4.34 (4H, t, *J* = 6.5 Hz, 2 × N-CH_2_), 4.01 (4H, ps, 2 × O-CH_2_), 3.04 (4H, t, *J* = 6.8 Hz, 2 × C*H*_2_-C(=C)N), ), 2.72 (4H, t, *J* = 6.6 Hz, 2 × C*H*_2_-C=O), 2.40 (2H, dd, *J* = 3.7 Hz, *J* = 16.8 Hz, H-4_eq_, H-4′_eq_), 2.11–2.02 (2H, m, H-4_ax_, H-4′_ax_), 1.95 (12H, s, CH_3_-19,19′,20,20’), 1.80 (2H, pt, *J* = 6.4 Hz, H-2_eq_, H-2′_eq_), 1.76–1.73 (8H, m, 4 × CH_2_), 1.71 (6H, s, CH_3_-18,18′), 1.58-1.52 (6H, m, H-2_ax_, H-2′_ax_, 2 x CH_2_), 1.42–1.33 (4H, m, 2 × CH_2_), 1.10, 1.06 (12H, 2s, CH_3_-16,16′,17,17′); ^13^C-NMR (CDCl_3_, 125 MHz) *δ* 182.4 (C, 2 × C-4c), 172.4 (C, 2 × *C*=O), 165.0 (C, 2 × C-7a), 163.9 (C, 2 × C-2c), 162.2 (C, 2 × C-5a), 157.8 (C, 2 × C-8c), 146.5 (C, 2 × triazole N*C*=C), 138.7 (CH, C-8, 8′), 137.6 (CH, C-12,12′), 137.9, 136.4, 135.5 (C, C-5,5′,6,6′,9,9′*), 132.6 (C-14,14′), 131.8 (CH, 2 × C-4b) 131.5 (CH, C-10,10’), 131.3 (C, 2 × C-1b), 130.1 (CH, C-15,15′), 129.0 (CH, 2 × C-3b), 126.3 (CH, 2 × C-2b), 125.5 (C, C-13,13′*), 125.2 (CH, C-7,7′), 124.8 (CH, C-11,11′), 121.0 (CH, 2 × triazole CH), 105.9 (CH, 2 × C-3c), 105.7 (C, 2 × C-4a), 98.6 (CH, 2 × C-6a), 93.1 (CH, 2 × C-8a), 68.6 (CH, C-3,3′), 68.3 (CH_2_, 2 × *C*H_2_-O), 50.0 (CH_2_, 2 × *C*H_2_-N), 44.0 (CH_2_, C-2,2′), 38.4 (CH_2_, C-4,4′), 36.7 (C, C-1,1′), 34.0 (CH_2_, 2 × *C*H_2_-CO), 30.1, 28.5 (CH_3_, CH_3_-16,16′,17,17′), 30.2, 28.8, 26.2, 25.5 (CH_2_, 2 × 4 CH_2_ linker), 21.5 (CH_3_, CH_3_-18,18′), 21.1 (CH_2_, 2 × *C*H_2_-C(=C)N), 12.8, 12.7 (CH_3_, CH_3_-19,19′,20,20’) *interchangeable; MS (MALDI-TOF, positive mode, with DHB matrix) *m*/*z* 1509.913 [M + Na]^+^; Anal. calcd. for C_92_H_106_N_6_O_12_: C 74.27, H 7.18, N 5.65 found C 74.13, H 7.83 N 5.09.

*3-(1-(6-((5-hydroxy-2-phenyl-4H-chromen-4-one-7-yl)oxy)hex-1-yl)-1,2,3-triazol-4-yl)-propanoyloxy)-3′-(4-pentynoyloxy)-β,β-carotene* (**23**): orange-red crystals (60%); mp 122–123 °C; UV (THF) *λ*_max_ 277, 429, 458, 486 nm; IR (KBr) ν*_max_* 3308, 2922, 1727, 1613 cm^−1^; ^1^H-NMR (CDCl_3_, 500 MHz) *δ* 12.71 (1H, s, OH), 7.88 (2H, d, *J* = 6.6 Hz, 2 × H-2b), 7.56–7.50 (3H, m, 2 × H-3b, H-4b), 7.35 (1H, s, triazole CH), 6.66 (1H, s, H-3c), 6.65-6.60 (4H, m, H-11,11′,15,15′), 6.48 (1H, d, *J* = 2.1 Hz, H-8a), 6.36 (2H, dd, *J* = 5.5 Hz, *J* = 14.7 Hz, H-12,12′), 6.35 (1H, s, H-6a), 6.25 (2H, pd, *J* = 7.4 Hz, H-14,14′), 6.15 (2H, dd, *J* = 6.4 Hz, *J* = 11.5 Hz, H-10,10’), 6.12–6.05 (4H, m, H-7,7′,8,8′), 5.13–5.03 (2H, m, H-3,3′), 4.34 (2H, t, *J* = 7.1 Hz, N-CH_2_), 4.02 (2H, t, *J* = 6.3 Hz, O-CH_2_), 3.05 (2H, t, *J* = 7.2 Hz, CH_2_-C(=C)N), 2.72 (4H, t, *J* = 7.3 Hz, CH_2_-C=O), 2.56–2.50 (4H, m, CH_2_ pentynoate), 2.47–2.37 (2H, m, H-4_eq_,4′_eq_), 2.15–2.06 (2H, m, H-4_ax_,4′_ax_), 1.98, 1.96, 1.95 (3s, 13H, HC≡C, CH_3_-19,19′,20,20’), 1.94–1.91 (2H, m, CH_2_-γ, overlayed with the singlet at 1.95 ppm), 1.84–1.76 (4H, m, H-2_eq_, H-2′_eq_, CH_2_-α), 1.72, 1.71 (6H, 2 s, CH_3_-18,18′), 1.61–1.50 (4H, m, H-2_ax_, H-2′_ax_, CH_2_-β), 1.44–1.38 (2H, m, CH_2_-δ), 1.11, 1.10, 1.07, 1.06 (12H, 4s, CH_3_-16,16′,17,17′); ^13^C NMR (CDCl_3_, 125 MHz) *δ* 182.4 (C, C-4c), 172.5, 171.5 (C, 2 × O*C*=O), 165.0 (C, C-7a), 164.0 (C, C-2c), 162.2 (C, C-5a), 157.8 (C, C-8c), 146.5 (C, triazole N*C*=C), 138.7 (CH, C-8,8′), 137.9, 136.5, 135.6, 135.5 (C, C-5,5′,6,6′,9,9′*), 137.7 (CH, C-12,12′), 132.6 (CH, C-14,14′), 131.8 (CH, C-4b), 131.5 (CH, C-10,10’), 131.4 (C, C-1b), 130.1 (CH, C-15,15′), 129.1 (CH, 2 × C-3b), 126.3 (CH, 2 × C-2b), 125.5 (C, C-13,13′*), 125.3, 125.2 (CH, C-7,7′), 124.9 (CH, C-11,11′), 121.0 (CH, triazole CH), 105.9 (CH, C-3c), 105.7 (C, C-4a), 98.6 (CH, C-6a), 93.1 (CH, C-8a), 82.6 (C, *C*≡CH), 68.9, 68.6 (CH, C-3,3′), 68.3 (CH_2_, CH_2_-O), 50.1 (CH_2_-N), 44.04, 44.03 (CH_2_, C-2,2′), 38.4 (CH_2_, C-4,4′), 36.72, 36.70 (C, C-1,1′), 34.0, 33.7 (CH_2_, 2 × CH_2_-CO), 30.2 (CH_2_, CH_2_-γ), 30.0, 28.53, 28.51 (CH_3_, CH_3_-16,16′,17,17′), 28.8 (CH_2_, CH_2_-α), 26.2 (CH_2_, CH_2_-δ), 25.5 (CH_2_, CH_2_-β), 21.5 (CH_3_, CH_3_-18,18′), 21.1 (CH_2_, *C*H_2_-C(=C)N), 14.4 (CH_2_, *C*H_2_-C≡CH), 12.8, 12.7 (CH_3_, CH_3_-19,19′,20,20’) *interchangeable; MS (MALDI-TOF, positive mode, with DHB matrix) *m*/*z* 1130.7 [M + Na]^+^; Anal. calcd. for C_71_H_85_N_3_O_8_: C 76.93, H 7.73, N 3.79 found C 76.32, H 7.45 N 3.50.

*3,3′-bis(1-(6-((5-hydroxy-2-phenyl-4H-chromen-4-one-7-yl)oxy)hex-1-yl)-1,2,3-triazol-4-yl)-propanoyloxy)-β,κ-carotene* (**24**): dark red crystals (88%); mp 88–90 °C; UV (THF) *λ*_max_ 243, 278, 476 nm; IR (KBr) ν*_max_* 3414, 1728, 1641, 1624, 1252 cm^−1^; ^1^H NMR (CDCl_3_, 500 MHz) *δ* 12.70 (2H, s, 2 × OH), 7.88 (4H, d, *J* = 6.9 Hz, 4 × H-2b), 7.52 (6H, pd, *J* = 7.6 Hz, 4 × H-3b, 2 × H-4b), 7.34 (3H, brs, 2 × triazole CH, H-8′), 6.71–6.46 (6H, m, H-10’,11,11′,12′,15,15′), 6.65 (2H, s, 2 × H-3c), 6.48 (2H, s, 2 x H-8a), 6.42-6.32 (3H, m, H-7′,12,14′), 6.34 (2H, s, 2 × H-6a), 6.25 (1H, d, *J* = 11.1 Hz, H-14), 6.15 (1H, d, *J* = 11.0 Hz, H-10), 6.12-6.07 (2H, m, H-7, H-8), 5.30-5.21 (m, 1H, H-3′), 5.11–5.02 (m, 1H, H-3), 4.33 (4H, pd, *J* = 4.9 Hz, 2 × N-CH_2_), 4.01 (4H, t, *J* = 5.6 Hz, 2 × O-CH_2_), 3.06–3.02 (4H, m, 2 × CH_2_-C(=C)N), 2.96 (1H, dd, *J* = 9.0 Hz, *J* = 14.8 Hz, H-4′α), 2.74-2.69 (4H, m, 2 × CH_2_-C=O), 2.40 (dd, 1H, *J* = 5.1 Hz, *J* = 16.9 Hz, H-4_eq_), 2.11–2.03 (2H, m, H-4_ax_, H-2′α), 1.98, 1.96, 1.94 (12H, 3s, CH_3_-19,19′,20,20’), 1.85–1.73 (6H, m, H-2_eq_, H-2′_β_, 2 CH_2_ linker), 1.71 (3H, s, CH_3_-18), 1.58–1.47 (6H, m, H-2_ax_, H-4′_β_, 2 CH_2_ linker), 1.43–1.35 (4H, m, 2 CH_2_ linker), 1.29 (3H, s, CH_3_-18′), 1.28–1.21 (4H, m, 2 CH_2_ linker, overlayed with the s at 1.29 ppm), 1.15, 1.10 (6H, 2s, CH_3_-16,17) 1.06, 0.84 (6H, 2s, CH_3_-16′,17′); ^13^C-NMR (CDCl_3_, 125 MHz) *δ* 202.4 (C, C-6′), 182.4 (C, 2 × C-4c), 172.5, 172.4 (C, 2 × O*C*=O), 165.0 (C, 2 × C-7a), 163.9 (C, 2 × C-2c), 162.2 (C, 2 × C-5a), 157.8 (C, 2 × C-8c), 147.1 (CH, C-8′), 146.5, 146.4 (C, 2 × triazole N*C*=C), 142.1, 140.9, 138.6, 137.5, 135.3, 132.4, 131.7, 129.7 (CH, C-8,10,10’,12,12′,14,14′,15,15′), 137.9, 137.6, 136.0, 135.9, 133.6 (C, C-6,9,9′,13,13′), 131.8 (CH, 2 × C-4b), 129.1 (CH, 4 × 3b), 128.9 (C, 2 C-1b, partially overlayed with C-3b), 126.3 (CH, 4 × C-2b), 125.6 (C, C-5), 125.5, 124.0, 121.1, 121.0 (CH, 2 × triazole CH, C-7,11,11′), 120.6 (CH, C-7′), 105.9 (CH, 2 × C-3c), 105.7 (C, 2 × C-4a), 98.6 (CH, 2 × C-6a), 93.1 (CH, 2 × C-8a), 73.7 (CH, C-3′), 68.6 (CH, C-3), 68.3 (CH_2_, 2 × CH_2_-O), 58.6 (C, C-5′), 50.1 (CH_2_, 2 × CH_2_-N), 47.6 (CH_2_, C-2′), 44.0 (CH_2_, C-2), 43.7 (CH_2_, C-4′), 42.3 (CH_2_, C-4), 38.4, 36.7 (C, C-1,1′), 34.0, 33.9 (CH_2_, 2 × *C*H_2_-CO), 30.2, 28.7, 26.2, 25.5 (CH_2_, 8 × CH_2_), 30.2, 28.5, 25.6, 24.8 (CH_3_, CH_3_-16,16′,17,17′), 21.5, 20.7 (CH_3_, CH_3_-18,18′), 21.1 (CH_2_, 2x *C*H_2_-C(=C)N), 12.9, 12.8, 12.7, (CH_3_, CH_3_-19,19′,20,20’); MS (MALDI-TOF, positive mode, with DHB matrix) *m*/*z* 1526.1 [M + Na]^+^; Anal. calcd. for C_92_H_106_N_6_O_13_: C 73.48, H 7.10, N 5.59 found C 73.05, H 6.98 N 55.46.

*3-(1-(6-((5-hydroxy-2-phenyl-4H-chromen-4-one-7-yl)oxy)hex-1-yl)-1,2,3-triazol-4-yl)-propanoyloxy)-3′-(4-pentynoyloxy)-β,κ-carotene and 3-(4-pentynoyloxy)-3′-(1-(6-((5-hydroxy-2-phenyl-4H-chromen-4-one-7-yl)oxy)hex-1-yl)-1,2,3-triazol-4-yl)-propanoyloxy)-β,κ-carotene* (**25**): dark red crystals (21%); mp 139–140 °C; UV (THF) *λ*_max_ 242, 288, 476 nm; IR (KBr) ν*_max_* 3313, 1609, 1663, 2924 cm^−1^; ^1^H NMR (CDCl_3_, 500 MHz) *δ* 12.71 (H, s, OH), 7.89 (2H, d, *J* = 6.9 Hz, 2 × H-2b), 7.55–7.51 (3H, m, 2 × H-3b, H-4b), 7.35–7.33 (2H, m, triazole CH, H-8′), 6.67 (1H, s, H-3a), 6.73–6.51 (6H, m, H-10’,11,11′,12′,15,15′), 6.49 (1H, ps, H-8a), 6.45–6.33 (3H, m, H-7′,12,14′), 6.35 (1H, s, H-6a), 6.26 (1H, d, *J* = 11.1 Hz, H-14), 6.18–6.16 (3H, m, H-7,8,10), 5.31–5.24 (m, 1H, H-3′), 5.14–5.05 (m, 1H, H-3), 4.36–4.32 (2H, m, N-CH_2_), 4.03 (2H, t, *J* = 6.0 Hz, O-CH_2_), 3.07–3.03 (2H, m, CH_2_-C(=C)N), 2.98 (1H, dd, *J* = 5.6 Hz, *J* = 14.3 Hz, H-4′α), 2.75–2.70 (2H, m, CH_2_-C=O), 2.55–2.51 (2H, m, 2 × CH_2_ pentynoate), 2.43–2.38 (m, 1H, H-4_eq_), 2.15–2.06 (2H, m, H-2′_α_, H-4_ax_), 1.99, 1.98, 1.97, 1.95 (13H, 4s, CH≡C, CH_3_-19,19′,20,20’), 1.83–1.78 (4H, m, H-2_eq_, H-2′_β_, CH_2_), 1.73, 1.72 (3H, 2 s, CH_3_-18), 1.62–1.50 (6H, m, 2 × CH_2_ H-2_ax_, H-4′_β_), 1.45–1.38 (4H, m, 2 CH_2_), 1.29, 1.30 (3H, 2 s, CH_3_-18′), 1.19, 1.16, 1.12, 1.11 (6H, 4 s, CH_3_-16′,17′), 1.08, 1.07, 0.89, 0.85 (2 s, CH_3_-16,17, overlaid with the signal of grease). The methyl groups of 16,16′,17,17′,18 and 18′ give separate signals for the two regioisomers; ^13^C NMR (CDCl_3_, 125 MHz) *δ* 202.4 (C, C-6′), 182.4 (C, C-4c), 172.5, 172.4 (C, 2 × O*C*=O), 165.0 (C, C-7a), 163.9 (C, C-2c), 162.2 (C, C-5a), 157.8 (C, C-8c), 147.1 (CH, C-8′), 146.5 (C, triazole N*C*=C), 142.1, 140.9, 138.6, 137.5, 135.3, 132.4, 131.8, 131.4, 129.7 (CH, C-8,10,10’,12,12′,14,14′,15,15′), 137.9, 137.6, 136.0, 135.9, 133.6 (C, C-6,9,9′,13,13′), 131.8 (CH, C-4b), 129.1 (CH, 2 × 3b), 128.3 (C, C-1b), 126.3 (CH, 2 × C-2b), 125.6 (C, C-5), 125.5, 124.1, 121.0, 121.0 (CH, triazole CH, C-7,11,11′), 120.6 (CH, C-7′), 105.9 (CH, C-3c), 105.7 (C, C-4a), 98.6 (CH, C-6a), 93.1 (CH, C-8a), 82.5 (C, *C*≡CH), 74.0, 73.8 (CH, C-3′), 69.0 (CH, H*C*≡C), 68.9, 68.6 (CH, C-3), 68.3 (CH_2_, CH_2_-O), 58.6 (C, C-5′), 50.0 (CH_2_, CH_2_-N), 47.7, 47.6 (CH_2_, C-2′), 44.1 (CH_2_, C-2), 43.7 (CH_2_, C-4′), 42.3 (CH_2_, C-4), 38.5, 36.7 (C, C-1,1′), 34.1, 34.0, 33.8, 33.7 (CH_2_, 2 × *C*H_2_-CO), 30.2, 28.8, 26.2, 25.5 (CH_2_, 4 × CH_2_), 30.2, 28.5, 25.6, 24.8 (CH_3_, CH_3_-16,16′,17,17′), 21.5, 20.7 (CH_3_, CH_3_-18,18′), 21.1 (CH_2_, *C*H_2_-C(=C)N), 14.5 (*C*H_2_-C≡C), 12.9, 12.8, 12.75, 12.70, (CH_3_, CH_3_-19,19′,20,20’); MS (MALDI-TOF, positive mode, with DHB matrix) *m*/*z* 1124.3 [M + H]^+^.

## Figures and Tables

**Figure 1 molecules-25-00636-f001:**
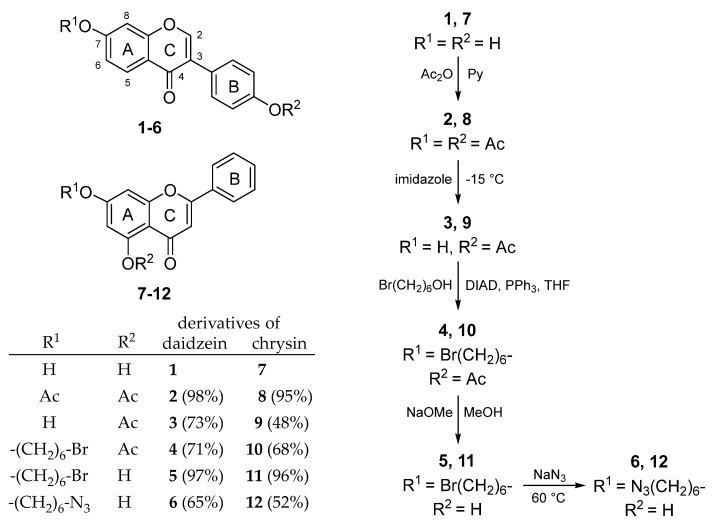
Synthesis of 7-azidohexyl ethers of daidzein and chrysin.

**Figure 2 molecules-25-00636-f002:**
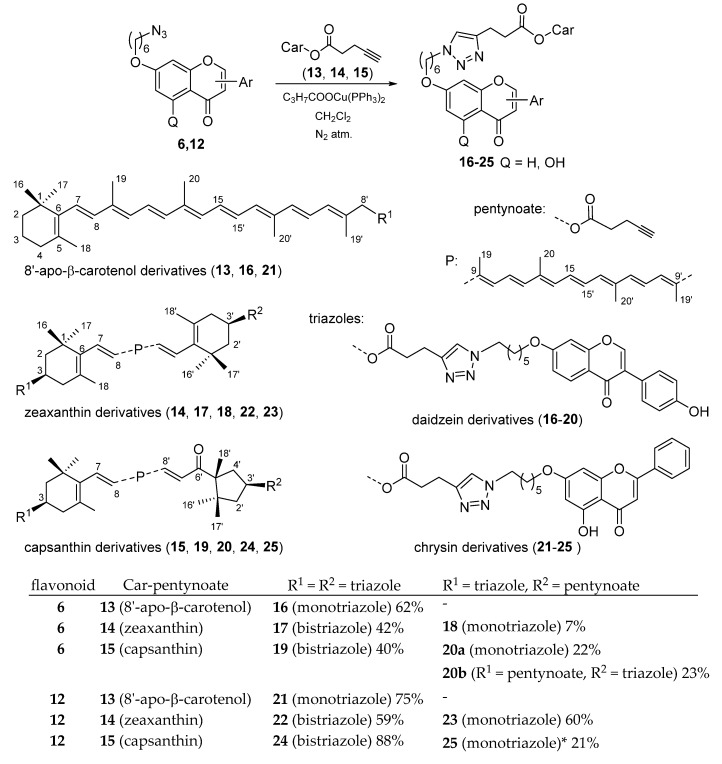
Coupling of 7-azidohexyl ethers of daidzein and chrysin with carotenoid pentynoates by azide-alkyne click-reaction. * In compound **25**, the R^1^ and R^2^ groups are interchangeable.

**Figure 3 molecules-25-00636-f003:**
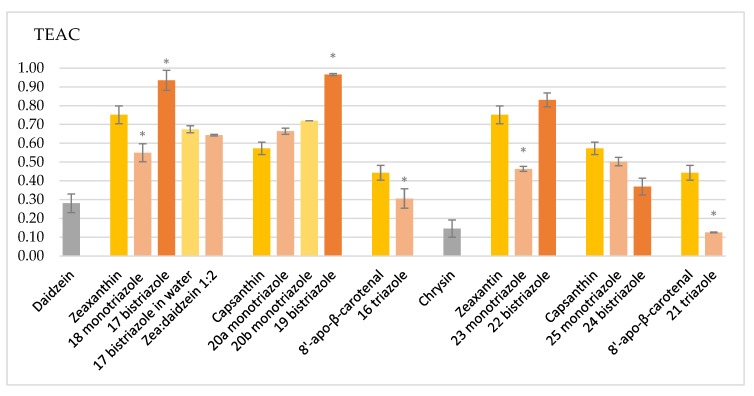
TEAC values of the parent flavonoids and carotenoids, as well as those of their conjugates. The error bars show the standard deviation. * indicates significant differences compared to the parent carotenoid (*p* < 0.05).

**Figure 4 molecules-25-00636-f004:**
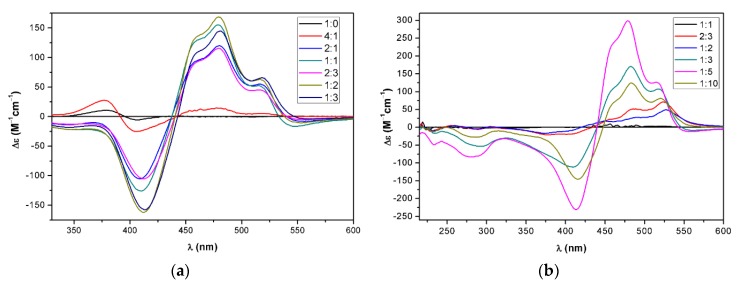
(**a**) The ECD spectra of bistriazole daidzein-zeaxanthin conjugate **17** in acetone/water mixtures 1:0–1:3 (**b**) in THF/water mixtures 1:1–1:10.

**Figure 5 molecules-25-00636-f005:**
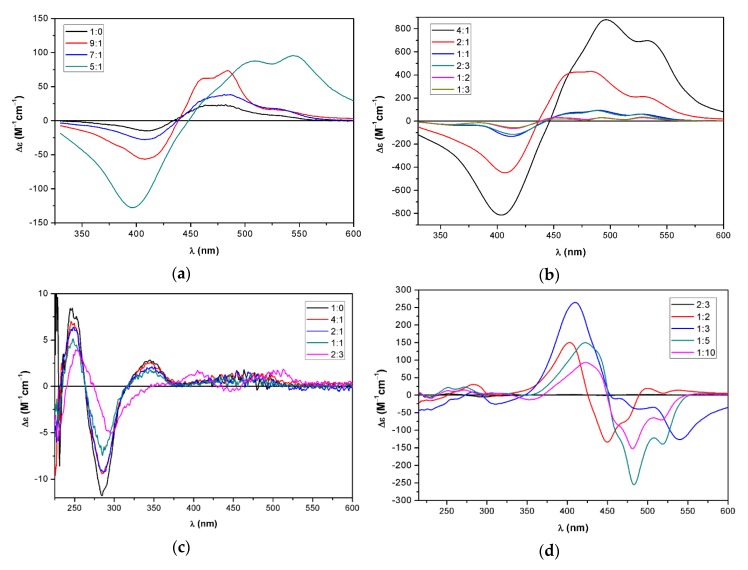
ECD spectra of bistriazole chrysin–zeaxanthin conjugate **22** in (**a**) acetone/water 1:0–5:1; (**b**) acetone/water 4:1–1:3 (**c**) THF/water 1:0–2:3. (**d**) THF/water 2:3–1:10.

**Figure 6 molecules-25-00636-f006:**
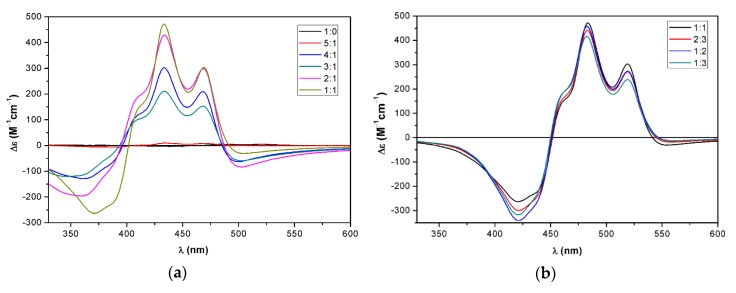
ECD spectra of monotriazole chrysin–zeaxanthin conjugate **23** in (**a**) acetone/water 1:0–1:1; (**b**) acetone/water 1:1–1:3.

**Figure 7 molecules-25-00636-f007:**
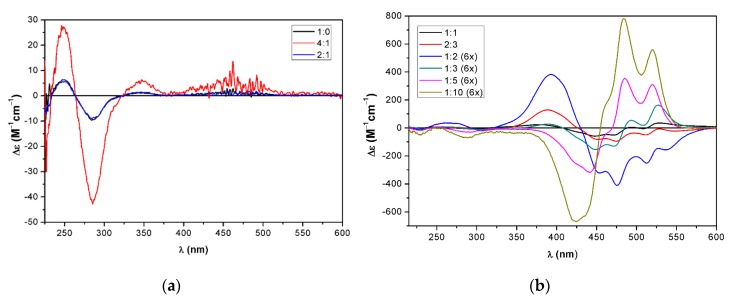
ECD spectra of monotriazole chrysin–zeaxanthin conjugate **23** in (**a**) THF/water 1:0–2:1. (**b**) THF/water 1:1–1:10.

**Figure 8 molecules-25-00636-f008:**
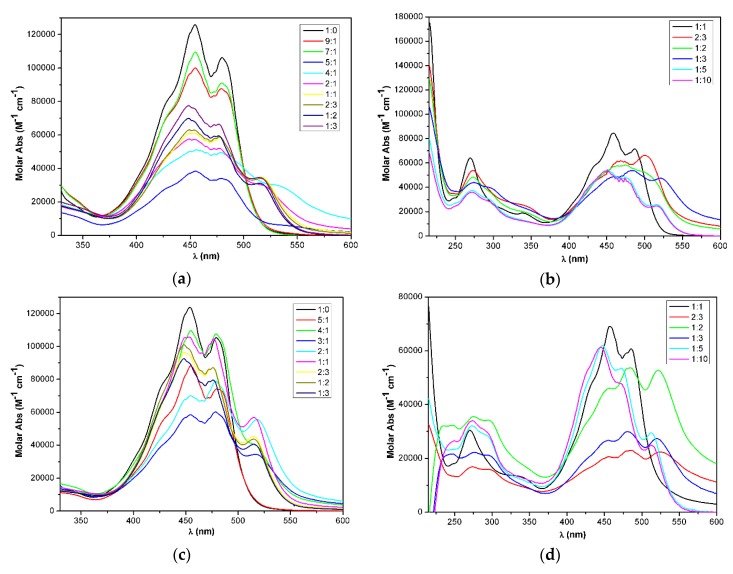
UV-vis spectra of chrysin–zeaxanthin conjugates of (**a**) the bistriazole **22** in acetone/water 1:0–1:3; (**b**) bistriazole **22** in THF/water 1:1–1:10; (**c**) monotriazole **23** in acetone/water 1:0–1:3; (**d**) monotriazole **23** in THF/water 1:0–1:1.

**Figure 9 molecules-25-00636-f009:**
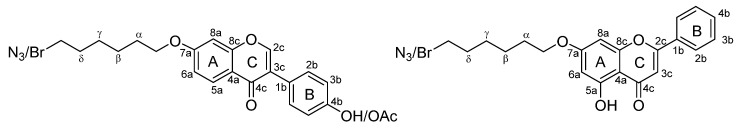
Numbering of the flavonoid derivatives.

**Table 1 molecules-25-00636-t001:** Chemical shifts of H-3 and H-3′ in capsanthin dipentynoate (**15**), in bistriazole **19**, and in monotriazoles **20a** and **20b**.

Compound	*δ* H-3 m (ppm)	*δ* H-3′ m (ppm)
capsanthin dipentynoate **15**	5.16–5.10	5.33–5.29
bistriazole **19**	5.09–5.03	5.26–5.22
monotriazole **20a**	5.13–5.03	5.31–5.25
monotriazole **20b**	5.14–5.07	5.27–5.22

**Table 2 molecules-25-00636-t002:** The TEAC values, standard deviations, and variance by the ANOVA method.

Compound	TEAC	SD%	Variance	Compound	TEAC	SD%	Variance
Daidzein	0.2804	4.9981	0.0070	Chrysin	0.1458	4.5581	0.0132
Zeaxanthin	0.7514	4.7639	0.0208	Zeaxanthin	0.7514	4.7639	0.0208
**18** monotriazole	0.5496	4.8011	0.0035	**23** monotriazole	0.4636	1.3732	0.0066
**17** bistriazole	0.9353	5.3524	0.1123	**22** bistriazole	0.8313	3.7065	0.0064
**17** bistriazole in water	0.6744	1.8747	0.0516				
Zea:daidzein 1:2 mixt.	0.6434	0.4262	0.0316				
Capsanthin	0.5728	3.2883	0.0095	Capsanthin	0.5728	3.2883	0.0095
**20a** monotriazole	0.6643	1.6416	0.0095	**25** monotriazole	0.5026	2.1947	0.0091
**20b** monotriazole	0.7199	0.0446	0.0033	**24** bistriazole	0.3698	4.4718	0.0242
**19** bistriazole	0.9663	0.5469	0.0463				
8′-apo-β-carotenal	0.4433	3.9429	0.0068	8′-apo-β-car.	0.4433	3.9429	0.0068
**16** triazole	0.3057	5.1589	0.0157	**21** triazole	0.1256	0.2122	0.0284

**Table 3 molecules-25-00636-t003:** Statistical analysis of the TEAC values.

Comparison	F Test	T Test	Comparison	F Test	T Test
Daidzein:**18**	74.28%	0.04%	Daidzein:**16**	27.71%	94.39%
Daidzein:**17**	0.32%	4.18%	8′-apo-β-car:**16**	67.89%	4.93%
Daidzein:**17** in water	45.15%	0.00%			
Daidzein:Zea-daid. mixt.	9.80%	0.00%	Chrysin:**23**	23.74%	3.63%
Zeaxanthin:**18**	30.03%	4.99%	Chrysin:**22**	65.04%	0.86%
Zeaxanthin:**17**	6.57%	1.05%	Zeaxanthin:**23**	52.52%	2.10%
Zeaxanthin:**17** in water	29.05%	15.51%	Zeaxanthin:**22**	50.99%	27.87%
Zeaxanthin: Zea-daid. mixt.	55.19%	74.42%			
			Chrysin:**25**	81.78%	0.67%
Daidzein:**20a**	62.55%	0.51%	Chrysin:**24**	80.59%	3.33%
Daidzein:**20b**	41.90%	0.00%	Capsanthin:**25**	88.47%	7.40%
Daidzein:**19**	4.06%	0.00%	Capsanthin:**24**	34.33%	22.00%
Capsanthin:**20a**	86.17%	30.11%			
Capsanthin:**20b**	27.09%	22.39%	Chrysin:**21**	69.35%	32.28%
Capsanthin:**19**	13.29%	4.94%	8′-apo-β-car:**21**	41.10%	3.86%

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
