# Peer review of "Study on the Synthesis, Antioxidant Properties, and Self-Assembly of Carotenoid–Flavonoid Conjugates"

_molecules, 2020, doi:10.3390/molecules25030636_

Round 1

Reviewer 1 Report

The paper describes a synthetic procedure used for the covalent coupling of hydrophobic carotenoids with hydrophilic flavonoids such as daidzein and chrysin in order to obtain new amphipathic structures. Several interesting compounds were obtained and their syntheses are carefully described in the manuscript as well as the spectroscopic characterization of the new molecules. Antioxidant activity of the new compounds with respect to the parent flavonoids and carotenoids was also evaluated. In some cases synergistic enhancement for the antioxidant activities (diflavonoid hybrides of zeaxanthin and capsanthin) was observed. The interesting formation of J-aggregates was also observed and studied. The paper is well written, reports a considerable amount of synthetic work and deserves publication on Molecules. Further NMR investigations (solvent and concentration depending NMR shifts, DOSY experiments) could have been added (in case) to better elucidate the nature and occurrence of the aggregation phenomena.  

Minor points the position of daidzein and chrysin in the structure elucidation table of Fig 4 is misleading the names since correspond to compounds 1 and 7 only while the headings should indicate the general structures.

An English check should be performed. See for example in the introduction “Countless natural bioactive compounds has been isolated from food products….”.  Should be:  Countless natural bioactive compounds have been isolated from food products…..In the Abstract:  “hybrides” should be “hybrids”

Author Response

I would like to thank the reviewer’s work. We clarified the headings of the table according to the reviewer’s advice and checked the English again.

Reviewer 2 Report

In my opinion the paper is very well presented and very high level work
is there described. A novel method was worked out for the synthesis of novel
carotenoid-flavonoid conjugates and their antioxidant capacities
were measured as well.
Carotenoids and flavonoids are naturally occurring antioxidant
phytochemicals and novel properties may arise from the amphipatic
supramolecular assemblies preliminary described in this work.

In my opinion the paper can be published in the Molecules journal in its present form.

Author Response

We thank the reviewer to appreciate our work and to recommend it for publication.

Reviewer 3 Report

see pfd file

Author Response

I sincerely thank the reviewer for his thorough discussion of our manuscript, as well as his critical comments and advice. I will eminently take them into account in the future. We thank the reviewer to appreciate our work and to recommend it for publication.